# A molecular basis for the differential roles of Bub1 and BubR1 in the spindle assembly checkpoint

Katharina Overlack[1†], Ivana Primorac[1†], Mathijs Vleugel[2], Veronica Krenn[1], Stefano Maffini[1], Ingrid Hoffmann[1], Geert J P L Kops[3,4,6,7,8], Andrea Musacchio[1,5*]

[1]Department of Mechanistic Cell Biology, Max Planck Institute of Molecular Physiology, Dortmund, Germany; [2]Molecular Cancer Research, University Medical Center Utrecht, Utrecht, Netherlands; [3]Department of Molecular Cancer Research, University Medical Center Utrecht, Utrecht, Netherlands; [4]Department of Medical Oncology, University Medical Center Utrecht, Utrecht, Netherlands; [5]Centre for Medical Biotechnology, University Duisburg-Essen, Essen, Germany; [6]Cancer Genomics Netherlands, University Medical Center, Utrecht, Netherlands; [7]Department of Biology, Utrecht University, Utrecht, Netherlands; [8]Netherlands Proteomics Center, Utrecht, Netherlands

**\*For correspondence:** andrea. musacchio@mpi-dortmund.mpg. de

[†]These authors contributed equally to this work

**Competing interests:** The authors declare that no competing interests exist.

**Abstract** The spindle assembly checkpoint (SAC) monitors and promotes kinetochore–microtubule attachment during mitosis. Bub1 and BubR1, SAC components, originated from duplication of an ancestor gene. Subsequent sub-functionalization established subordination: Bub1, recruited first to kinetochores, promotes successive BubR1 recruitment. Because both Bub1 and BubR1 hetero-dimerize with Bub3, a targeting adaptor for phosphorylated kinetochores, the molecular basis for such sub-functionalization is unclear. We demonstrate that Bub1, but not BubR1, enhances binding of Bub3 to phosphorylated kinetochores. Grafting a short motif of Bub1 onto BubR1 promotes Bub1-independent kinetochore recruitment of BubR1. This gain-of-function BubR1 mutant cannot sustain a functional checkpoint. We demonstrate that kinetochore localization of BubR1 relies on direct hetero-dimerization with Bub1 at a pseudo-symmetric interface. This pseudo-symmetric interaction underpins a template–copy relationship crucial for kinetochore–microtubule attachment and SAC signaling. Our results illustrate how gene duplication and sub-functionalization shape the workings of an essential molecular network.

## Introduction

Bub1 and BubR1 are paralogous proteins involved in the spindle assembly checkpoint (SAC), a safety device that monitors the attachment of kinetochores to spindle microtubules and halts mitotic progression until completion of chromosome bi-orientation on the mitotic spindle (*Lara-Gonzalez et al., 2012*; *Foley and Kapoor, 2013*). Bub1 and BubR1 originated from a gene that was already present in the hypothetical last eukaryotic common ancestor (LECA) (*Suijkerbuijk et al., 2012a*). After speciation, up to nine distinct duplication events might have occurred, which invariably led to sub-functionalization of the resulting gene products (*Suijkerbuijk et al., 2012a*).

Human Bub1 and BubR1 are strongly conserved at the sequence and domain level (*Figure 1A*) but play complementary roles in the SAC. Bub1 becomes recruited to kinetochores in prometaphase to provide a platform for additional SAC proteins, including Mad1, Mad2, the BubR1/Bub3 complex, and Cdc20. Bub1 promotes the assembly of a subset of these proteins, Mad2, BubR1/Bub3, and Cdc20,

**eLife digest** The genetic material within our cells is arranged in structures called chromosomes. Before a cell divides it makes an accurate copy of all of its DNA. The genetic material then needs to be equally split so that both daughter cells have a complete set of chromosomes.

As the cell prepares to divide, each chromosome—consisting of two identical sister chromatids—lines up on a structure known as the spindle, which is made of filaments called microtubules. Cells have a sophisticated safety mechanism known as the spindle assembly checkpoint to ensure that chromosomes have time to correctly line up on the spindle before the cell can divide. Once this checkpoint is satisfied, the microtubules pull the sister chromatids apart so that each daughter cell receives one chromatid from each pair.

The microtubules attach to the chromosomes through a large protein complex known as the kinetochore that assembles on each sister chromatid. The spindle assembly checkpoint monitors the attachment of the kinetochores to the microtubules; and two proteins, called Bub1 and BubR1, play an essential role in this process. These proteins bind to another protein called Bub3 that is also part of the spindle assembly checkpoint. Although Bub1 and BubR1 are very similar, they do not appear to perform the same roles, but the precise molecular details of their differences remain unclear.

In this study, Overlack, Primorac et al. studied Bub1 and BubR1 in human cells. The experiments show that Bub1 can be recruited to kinetochores in the absence of BubR1, but BubR1 will only move to kinetochores when Bub1 is present. Furthermore, BubR1 needs to bind to Bub1 directly to move to the kinetochores. Overlack, Primorac et al. also identified a region in Bub1 that binds to Bub3, and which is considerably different in BubR1. When this region of Bub1 was grafted into BubR1, the resulting protein was able to bind kinetochores even in the absence of Bub1.

The genes that encode the Bub1 and BubR1 proteins originate from a single ancestor gene that was duplicated during evolution. Therefore, the findings of Overlack, Primorac et al. show how the duplication of a gene can be beneficial for cells by creating products that have different roles in cells.

into the SAC effector, the mitotic checkpoint complex (MCC), which targets the anaphase promoting complex/cyclosome (APC/C) to inhibit its ability to promote mitotic exit (*Lara-Gonzalez et al., 2012*; *Foley and Kapoor, 2013*). Bub1, not in itself a MCC subunit, likely catalyzes MCC assembly by aligning MCC subunits for a profitable interaction (discussed in *Overlack et al., 2014*). Bub1 kinase activity is not required for the SAC (*Sharp-Baker and Chen, 2001*; *Fernius and Hardwick, 2007*; *Perera et al., 2007*; *Klebig et al., 2009*) but contributes to stable kinetochore–microtubule attachments through phosphorylation of histone H2A on Thr120 and subsequent localization of Sgo1 to centromeres and kinetochores (*Kawashima et al., 2010*; *Wang et al., 2011*; *Caldas et al., 2013*). BubR1, on the other hand, evolved into an inactive pseudo-kinase (*Suijkerbuijk et al., 2012a*) and is a crucial subunit of the MCC (*Hardwick et al., 2000*; *Fraschini et al., 2001*; *Sudakin et al., 2001*). BubR1 contributes to the formation of stable kinetochore–microtubule attachments and checkpoint silencing through kinetochore co-recruitment of protein phosphatase 2A (PP2A) (*Suijkerbuijk et al., 2012b*; *Kruse et al., 2013*; *Xu et al., 2013*; *Espert et al., 2014*; *Nijenhuis et al., 2014*).

Bub1 and BubR1 have different kinetochore dynamics, likely reflecting their distinct functions in the SAC. Bub1 interacts stably with unattached kinetochores, in agreement with its function as a SAC recruitment platform, while BubR1 turns over rapidly ($t_{1/2} = 3$–20 s), likely reflecting its cycle of incorporation into MCC and its release into the cytosol as a soluble APC/C inhibitor (*Howell et al., 2004*; *Shah et al., 2004*). Besides different dynamics, another important difference is that kinetochore recruitment of Bub1 is independent of BubR1, while recruitment of BubR1 is strictly subordinate to Bub1 (*Millband and Hardwick, 2002*; *Gillett et al., 2004*; *Johnson et al., 2004*; *Perera et al., 2007*; *Logarinho et al., 2008*; *Klebig et al., 2009*). The molecular basis for these differences is unclear, because both Bub1 and BubR1 bind a kinetochore-targeting adaptor named Bub3. Bub3 is a 7-WD40 β-propeller that targets kinetochores by binding to phosphorylated Met-Glu-Leu-Thr$^P$ (MELT$^P$, where T$^P$ indicates phosphothreonine) repeats of the outer kinetochore subunit Knl1 (a.k.a. Casc5, Spc105, Spc7, AF15q14, and Blinkin) (*Kiyomitsu et al., 2007*; *Krenn et al., 2012, 2014*; *London et al., 2012*; *Shepperd et al., 2012*; *Yamagishi et al., 2012*; *Primorac et al., 2013*) (*Figure 1B*). Bub3 binds tightly to Bub1 and BubR1 via conserved segments known as Bub3-binding domain (B3BD) or GLEBS

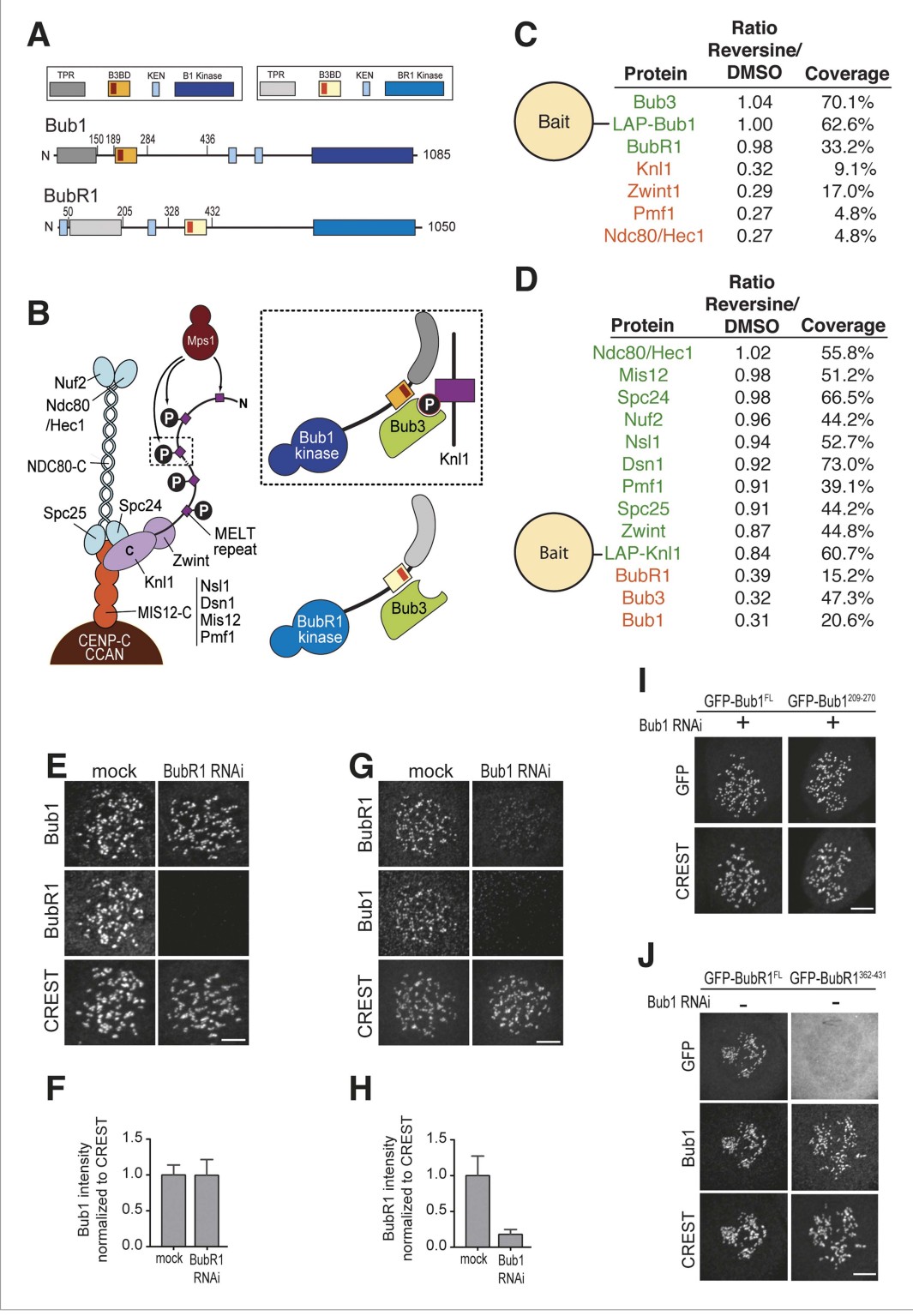

**Figure 1**. Mps1 and Bub1 are required for kinetochore localization of BubR1. (**A**) Similar domain organization of the homologous proteins Bub1 and BubR1. TPR—tetratrico peptide repeat, B3BD—Bub3 binding domain, B1—Bub1, BR1—BubR1. (**B**) Schematic depiction of the outer KT (KMN network). MELT repeats of Knl1 are phosphorylated by the checkpoint kinase Mps1 and recruit Bub1/Bub3 to KTs. It is not clear how BubR1 is recruited to KTs. (**C–D**) Quantitative IP-mass spectrometry analyses showing that the interaction of Bub1, BubR1, and Bub3 with KTs is significantly reduced upon inhibition of Mps1 with Reversine. Green- and red-labeled hits indicate respectively

*Figure 1. continued on next page*

*Figure 1. Continued*

proteins whose levels were not strongly affected or were strongly affected in the presence of Reversine. (**E** and **G**) Representative images of Flp-In T-REx cell lines in BubR1 (**E**) and Bub1 (**G**) RNAi, respectively, after treatment with nocodazole, showing that Bub1 is required for BubR KT localization. Scale bar: 10 μm. (**F** and **H**) Quantification of Bub1 and BubR1 KT levels, respectively, in cells treated as in panel **E** and **G**. The graph shows mean intensity, error bars indicate SD. The mean value for non-depleted cells is set to 1. (**I–J**) Representative images of stable Flp-In T-REx cell lines expressing the indicated GFP-Bub1 constructs (panel **I**) or HeLa cells transfected with the indicated GFP-BubR1 constructs (panel **J**) after treatment with nocodazole. The same images are also shown in *Figure 4A* and *Figure 5C*, and quantified in *Figures 4B and 5B*. Scale bar: 10 μm.
The following figure supplement is available for figure 1:

**Figure supplement 1**. RNAi quantification and schematic depiction of Bub1 and BubR1 constructs.

(*Taylor et al., 1998*; *Larsen et al., 2007*). By recognizing MELT[P], Bub3 co-recruits Bub1 to kinetochores in *Saccharomyces cerevisiae* (*Primorac et al., 2013*). In human cells, Bub3 is required for kinetochore recruitment of Bub1 and BubR1, and consistently the B3BDs of Bub1 and BubR1 are necessary, and in the case of Bub1 also sufficient, for kinetochore targeting of Bub1 and BubR1 (*Taylor et al., 1998*; *Logarinho et al., 2008*; *Malureanu et al., 2009*; *Elowe et al., 2010*; *Lara-Gonzalez et al., 2011*; *Krenn et al., 2012*). The subordination of BubR1 kinetochore recruitment to the presence of Bub1 suggests that Bub3 may operate differently when bound to Bub1 or BubR1. In this study, we set out to investigate the molecular basis of this phenomenon and its implications for spindle checkpoint signaling and kinetochore–microtubule attachment.

## Results

### Mps1 and Bub1 are required for kinetochore localization of BubR1

The SAC kinase Mps1 has been shown to phosphorylate MELT repeats of Knl1 to promote kinetochore recruitment of Bub1 and BubR1 (*Heinrich et al., 2012*; *London et al., 2012*; *Shepperd et al., 2012*; *Yamagishi et al., 2012*; *Primorac et al., 2013*; *Vleugel et al., 2013*; *Krenn et al., 2014*). We precipitated Bub1 or Knl1 (*Vleugel et al., 2013*) from mitotic lysates of HeLa cells treated with or without the Mps1 inhibitor Reversine (*Santaguida et al., 2010*). Quantitative mass spectrometry (see 'Materials and methods') of proteins associated with Bub1 or Knl1 confirmed the crucial role of Mps1, as we observed a strong suppression of the interaction of Bub1, BubR1, and Bub3 with kinetochores in the presence of Reversine (*Figure 1C–D*. Large deviations from a value of 1 for the Reversine/DMSO ratio indicate suppression of binding).

In HeLa cells treated with nocodazole, which depolymerizes microtubules and activates the SAC, Bub1 decorated kinetochores at essentially normal levels after the depletion of BubR1 (*Figure 1E*, quantified in *Figure 1F*. Quantifications of RNAi-based depletions are shown in *Figure 1—figure supplement 1A–B*). Conversely, BubR1 did not decorate kinetochores after Bub1 depletion (*Figure 1G–H*). These results confirm that BubR1 requires Bub1 for kinetochore recruitment, in line with previous studies (*Millband and Hardwick, 2002*; *Gillett et al., 2004*; *Johnson et al., 2004*; *Perera et al., 2007*; *Logarinho et al., 2008*; *Klebig et al., 2009*).

By monitoring the localization of a GFP-Bub1 reporter construct, we had previously demonstrated that Bub1[209-270], encompassing the B3BD, is the minimal Bub1 localization domain (*Taylor et al., 1998*; *Krenn et al., 2012*). Bub1[209–270] targeted kinetochores very efficiently even after the depletion of endogenous Bub1 (*Figure 1I*). We asked if an equivalent GFP reporter construct encompassing the B3BD of BubR1, BubR1[362–431], was also recruited to kinetochores. BubR1[362–431] was not recruited to kinetochores even in the presence of Bub1 (*Figure 1J*. Diagrams of Bub1 and BubR1 deletions used in this study are in *Figure 1—figure supplement 1C–D*). Thus, even if Bub1 and BubR1 share a related B3BD to interact with the same kinetochore-targeting subunit (Bub3) and interact in a phosphorylation-dependent manner with Knl1, the mechanisms of their kinetochore recruitment are different. This raises two crucial questions: (1) why is the B3BD region of Bub1 sufficient for kinetochore recruitment, while the equivalent region of BubR1 is not? And (2) if binding to Bub3 is not sufficient for robust

kinetochore recruitment of BubR1, how is BubR1 recruited to kinetochores? We will focus sequentially on these questions.

## The loop regions of Bub1 and BubR1 modulate the interaction of Bub3 with phosphorylated MELT motifs

To investigate if and how Bub1$^{209-270}$ and BubR1$^{362-431}$ modulate the binding affinity of Bub3 for the MELT$^P$ repeats of Knl1, we immobilized on amylose beads a fusion of maltose-binding protein (MBP) with residues 138–168 of Knl1, a region containing a single and functional MELT repeat (the most N-terminal, and therefore called MELT1; *Krenn et al., 2014*). We treated MBP-Knl1$^{MELT1}$ with or without Mps1 kinase. Next, we incubated MBP-Knl1$^{MELT1}$ with Bub3, Bub1$^{209-270}$/Bub3, or BubR1$^{362-431}$/Bub3 and visualized bound proteins by Western blotting. Bub3 in isolation did not bind MBP-Knl1$^{MELT1}$, in agreement with our previous data (*Krenn et al., 2014*). The B3BD of Bub1 strongly enhanced binding of Bub3 to phosphorylated MBP-Knl1$^{MELT1}$ but not to unphosphorylated MBP-Knl1$^{MELT1}$, while the B3BD of BubR1 had a negligible effect (*Figure 2A*). These results in vitro correlate with the ability of the equivalent B3BD to support (or not) kinetochore recruitment in cells (*Figure 1I–J*).

Our previous structural and biochemical characterization of the Bub1$^{B3BD}$ /Bub3/MELT$^P$ ternary complex of *S. cerevisiae* demonstrated that while Bub3 carries most of the crucial (and evolutionarily conserved) residues involved in high-affinity binding to MELT$^P$, a short segment of Bub1, the 'loop', contributes to the binding affinity (*Primorac et al., 2013*). The 'loop' region of Bub1 or BubR1 is between strands β1 and β2 (*Figure 2B–D*) and precedes the highly conserved core of the B3BD. The loop abuts the binding site for the MELT$^P$ peptide and is therefore ideally positioned to modulate the binding affinity of Bub3 for MELT$^P$ (*Figure 2C–D*).

Because the loops of Bub1 and BubR1 have quite divergent sequences (*Figure 2B*), we tested their role in modulating the binding affinity of Bub1$^{209-270}$/Bub3 or BubR1$^{362-431}$/Bub3 for immobilized MBP-Knl1$^{MELT1}$ (see *Figure 2A*). We swapped the loop regions of Bub1 and BubR1 as schematized in *Figure 2E*. Recombinant versions of the chimeric mutants Bub1$^{209-270/BR1-loop}$ and BubR1$^{362-431/B1-loop}$ were co-expressed with Bub3. Both the wild-type and chimeric constructs interacted with apparently similar affinity with Bub3, excluding gross structural perturbations (*Figure 2—figure supplement 1*). We then tested the ability of the recombinant constructs to interact with immobilized MBP-Knl1$^{MELT1}$. Bub1$^{209-270}$ bound tightly and in an Mps1-phosphorylation-dependent manner to MBP-Knl1$^{MELT1}$ (*Figure 2F*), while Bub1$^{209-270/BR1-loop}$ bound weakly. Conversely, BubR1$^{362-431/B1-loop}$ bound more strongly to phosphorylated MBP-Knl1$^{MELT1}$ than did BubR1$^{362-431}$ (*Figure 2F*). These results demonstrate a crucial role of the loop region of Bub1 in the recognition of a phosphorylated MELT repeat.

IP experiments from stable cell lines expressing GFP fusions of the wild-type or loop-swap mutants in the context of full-length Bub1 or BubR1 recapitulated the results in vitro (*Figure 2G–H*). Loop swap mutants interacted with Bub3 as efficiently as wild-type counterparts, but the interactions that Bub3 mediates, most notably those with the Knl1 receptor and its associated partner Mis12, were strongly enhanced when the Bub1 loop region was grafted onto BubR1.

## Behavior of the 'loop swap' mutants in HeLa cells

We then tested the localization behavior of these mutants. Kinetochore localization of GFP-Bub1$^{BR1-loop}$ was weaker than that of wild-type Bub1 (*Figure 3A–B*), whereas kinetochore localization of GFP-BubR1$^{B1-loop}$ was stronger than that of unmodified BubR1 (*Figure 3C–D*). Indeed, GFP-BubR1$^{B1-loop}$ decorated kinetochores even after the depletion of Bub1. Thus, when grafted onto BubR1, the Bub1 loop region is sufficient to confer upon BubR1 the ability to target kinetochores in a Bub1-independent fashion (*Figure 3E–F*). Although the experiments in vitro were carried out with a single MELT repeat, MELT1, they are likely to reflect the binding behavior of Bub1 and BubR1 to endogenous Knl1, which may contain up to 19 MELT repeats. Kinetochore localization of GFP-BubR1$^{B1-loop}$ in cells depleted of Bub1 was inhibited by addition of Reversine, indicating a dependence on Mps1 and MELT$^P$ sequences (*Figure 3G–H* and *Figure 3—figure supplement 1*, panel A). Expression of BubR1$^{B1-loop}$ did not overtly affect the kinetochore levels of endogenous Bub1 (*Figure 3—figure supplement 1*, panel B), suggesting that the multiple MELT$^P$ sequences of Knl1 are not saturated with endogenous Bub1 even in the strong checkpoint-activating conditions of our assay.

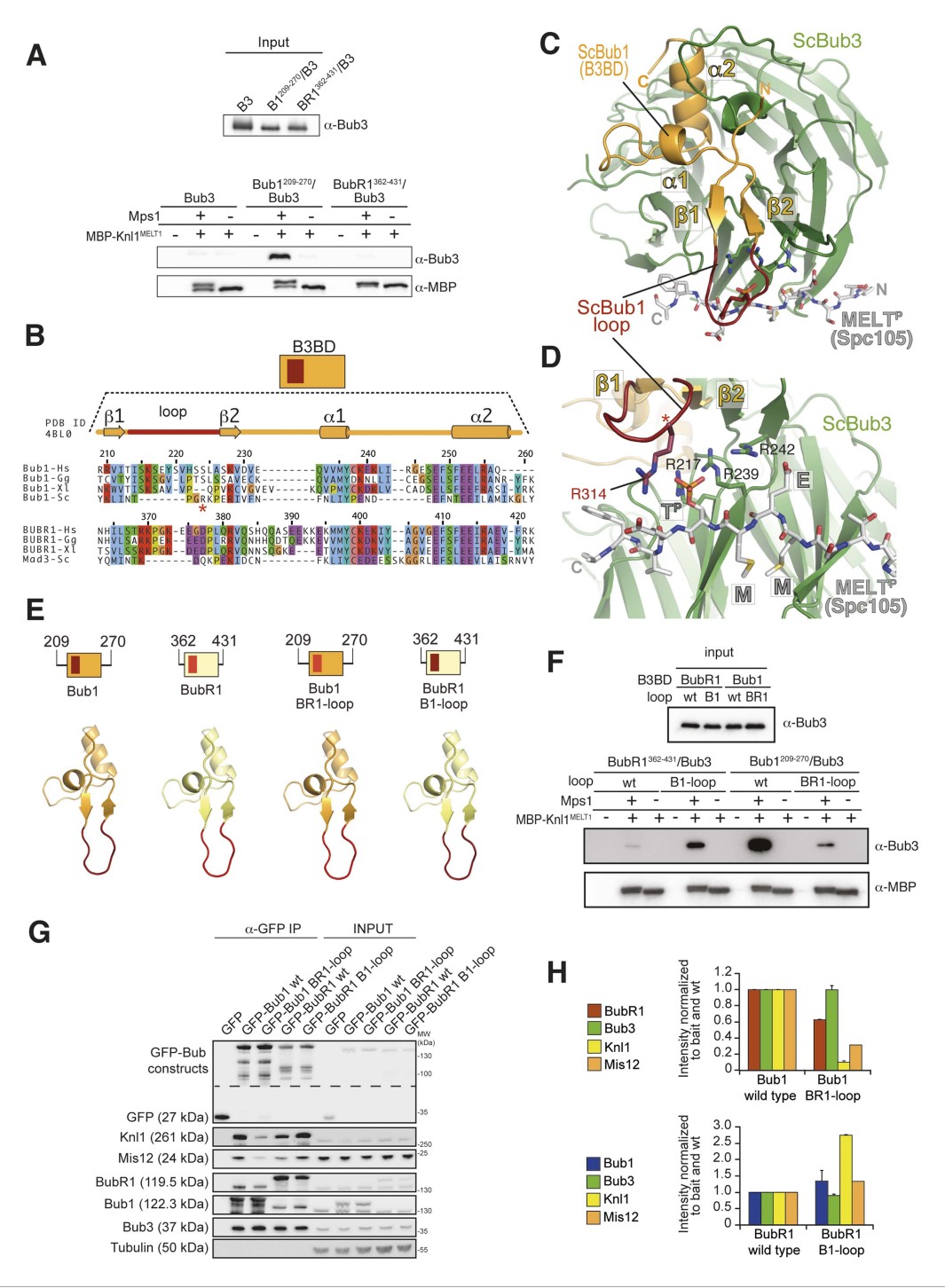

**Figure 2**. The loop regions of Bub1 and BubR1 modulate the interaction of Bub3 with phosphorylated MELT motifs. (**A**) Recombinant Bub3, Bub1²⁰⁹⁻²⁷⁰/Bub3 and BubR1³⁶²⁻⁴³¹/Bub3 were incubated with immobilized MBP-Knl1^MELT1 (residues 138–168 of human Knl1) prephosphorylated with Mps1 (+) or unphosphorylated (−). Empty lanes (−) demonstrate lack of background binding to empty beads. wt, wild type; B3, Bub3; B1, Bub1; BR1, BubR1. (**B**) Multiple sequence alignments of the Bub3 binding domains (B3BD) of human (*Homo sapiens*, Hs), chicken (*Gallus gallus*, gg), frog (*Xenopus laevis*, Xl), and budding yeast (*Saccharomyces cerevisiae*, Sc) Bub1 and BubR1s. Mad3 is the budding yeast BubR1 homolog. ScBub1^R314 (red asterisk) directly contributes to the interaction with the MELT^P peptide. The different Bub1 and BubR1 sequences were aligned manually on the basis of the crystal structures of the B3BDs of Mad3 and Bub1 in complex with Bub3 (*Larsen et al., 2007*; *Primorac et al., 2013*). *Figure 2. continued on next page*

*Figure 2. Continued*

(**C**) Crystal structure of the ScBub1$^{289–359}$-Bub3-MELT$^P$ ternary complex (*Primorac et al., 2013*). N and C indicate the N- and C-terminus, respectively. (**D**) Close-up of the MELT$^P$ binding site indicating the role of ScBub1$^{R314}$ in MELT$^P$ binding. (**E**) Schematic depiction of short Bub1 and BubR1 'loop swap' constructs, containing the loop (different shades of red) followed by the Bub3-binding domain (different shades of yellow). (**F**) Recombinant BubR1$^{362–431}$/Bub3 with its own loop (wt) or with the Bub1 loop (B1) and recombinant Bub1$^{209–270}$/Bub3 with its own loop (wt) or with the BubR1 loop (BR1) were incubated with immobilized MBP-Knl1$^{MELT1}$ prephosphorylated with Mps1 (+) or unphosphorylated (–) as in panel **A**. Empty lanes (–) demonstrate lack of background binding to empty beads. (**G**) Western Blot of immunoprecipitates (IP) from mitotic Flp-In T-REx cell lines expressing the indicated GFP-Bub1 and GFP-BubR1 constructs showing the influence of the loop on the ability to pull down the KT-components Knl1 and Mis12. Tubulin was used as loading control. (**H**) Quantification of the Western blot in (**G**). In the upper graph, the amounts of co-precipitating BubR1, Bub3, Knl1, and Mis12 were normalized to the amount of GFP-Bub1 bait present in the IP. In the lower graph, the amounts of co-precipitating Bub1, Bub3, Knl1, and Mis12 were normalized to the amount of GFP-BubR1 bait. Values for GFP-Bub1 wt and GFP-BubR1 wt, respectively are set to 1. The graphs show mean intensity of two independent experiments (for Mis12 only one). Error bars represent SD.

The following figure supplement is available for figure 2:

**Figure supplement 1**. Validation of recombinant 'loop swap' mutants.

---

Next, we asked if GFP-BubR1$^{B1–loop}$ was able to replace the SAC function of BubR1. In cells depleted of endogenous BubR1, wild-type BubR1 restored SAC function to a high degree but GFP-BubR1$^{B1–loop}$ failed to do so (*Figure 3I*). Comparison of immune-precipitates of GFP-BubR1 and GFP-BubR1$^{B1–loop}$ from nocodazole-treated mitotic cells showed much less of the latter associated with MCC and APC/C subunits (*Figure 3J*, quantified in *Figure 3—figure supplement 1*, panel C). Similarly, immune-precipitation (IP) of the APC/C demonstrated association of GFP-BubR1, but not GFP-BubR1$^{B1–loop}$, with the APC/C (*Figure 3K*, quantified in *Figure 3—figure supplement 1*, panel D). These results demonstrate that sequence divergence in the short loop region of Bub1 and BubR1 has strong functional consequences. In Bub1, the loop enhances the ability of Bub3 to recognize MELT$^P$ sequences of Knl1. In BubR1, the precise role of the loop is unknown, but the results in *Figure 3* suggest to us a specific role in MCC assembly or in the interaction with the APC/C, as discussed more thoroughly below.

## A minimal BubR1-binding region of Bub1

The second question, how BubR1 is recruited to kinetochores, has three distinct facets. First, the segment of Bub1 required for kinetochore recruitment of BubR1 should be identified. Second, the segment of BubR1 required for its own kinetochore recruitment should be identified. Third, it should be established if Bub1 and BubR1 interact directly and if kinetochore proteins other than Bub1 play additional roles in BubR1 recruitment.

To investigate the role of Bub1 in BubR1 recruitment, we created GFP fusions of several deletion mutants of Bub1 (*Figure 1—figure supplement 1C–D*) and tested concomitantly their kinetochore localization and BubR1 recruitment (*Figure 4A*). Bub1$^{209–270}$, encompassing the B3BD and localizing robustly to kinetochores, did not recruit BubR1 in cells depleted of endogenous Bub1 (*Figure 4A–B*). A construct containing the tetratrico peptide repeats (TPRs) and the B3BD of Bub1, Bub1$^{1–284}$, also targeted kinetochores but did not recruit BubR1 (*Figure 4A–B*). On the contrary, a segment (Bub1$^{209–409}$) consisting of the B3BD and a ~140-residue C-terminal extension (CTE), targeted kinetochores and promoted robust kinetochore localization of BubR1 (*Figure 4A–B*). Thus, Bub1 does not require the CTE (residues 271–409) for its own kinetochore recruitment (the B3BD, residues 209–270, is sufficient) but requires it to recruit BubR1. A deletion mutant lacking the CTE (Bub1$^{Δ271–409}$) targeted kinetochores efficiently but failed to recruit BubR1 (*Figure 4C–D*).

In IP experiments both in the presence and absence of endogenous Bub1, GFP-Bub1$^{209–409}$ interacted with BubR1 at levels that were only modestly lower than those of full-length GFP-Bub1 (*Figure 4E–F*). Conversely, GFP-Bub1$^{209–270}$ did not interact with BubR1, in agreement with the inability of this construct to promote BubR1 localization. In this context, it should be noted that the

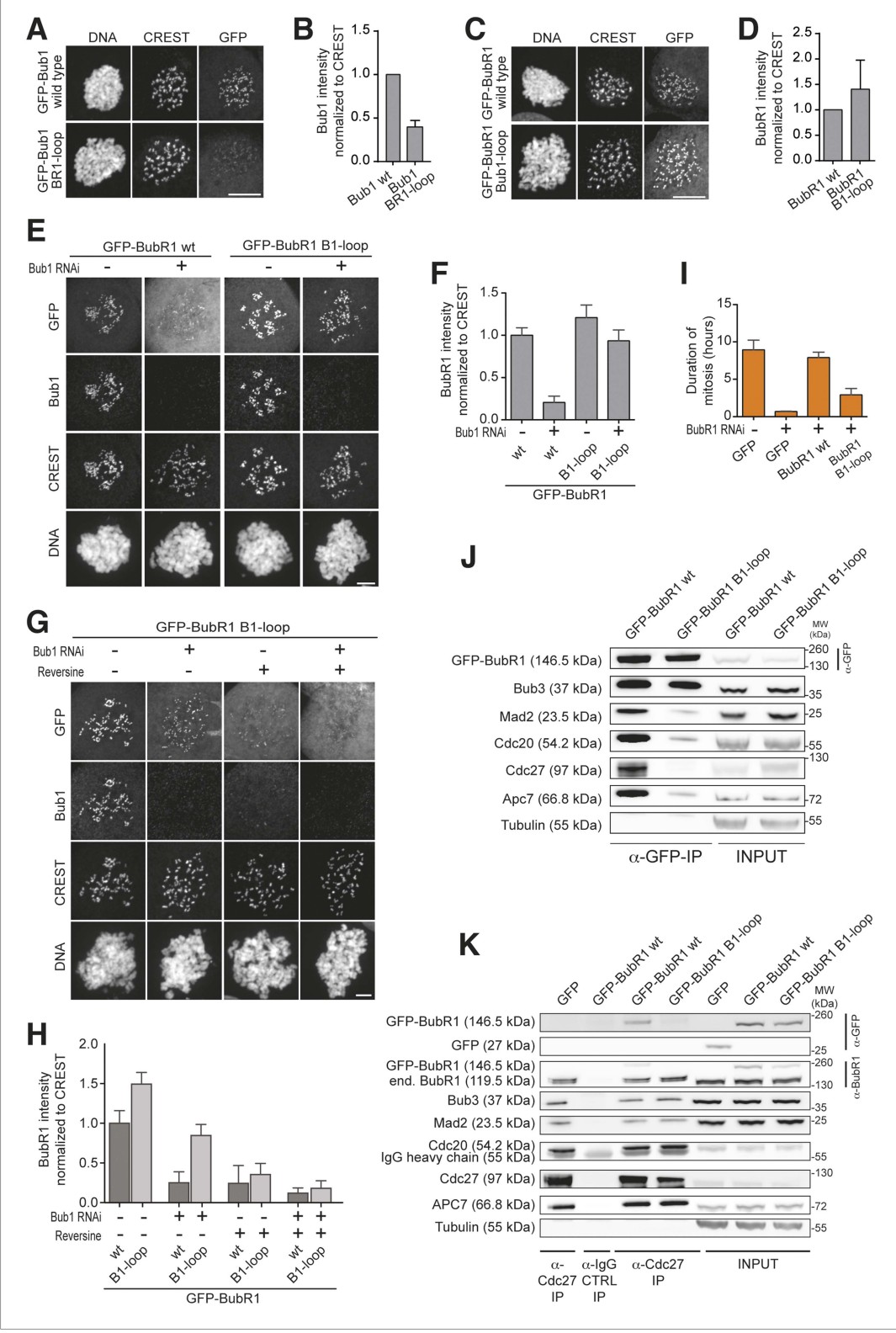

**Figure 3**. Behavior of the 'loop swap' mutants in HeLa cells. (**A**) Representative images of stable Flp-In T-REx cells expressing either GFP-Bub1 wild type (wt) or the loop mutant showing that the BR1-loop impairs KT localization. Scale bar: 10 µm. (**B**) Quantification of Bub1 KT levels in cells treated as in panel **A**. The graph shows mean intensity from three independent experiments. Error bar represents SEM. Values for Bub1 wt are set to 1. (**C**) Representative

*Figure 3. continued on next page*

*Figure 3. Continued*

images of stable Flp-In T-REx cells expressing either GFP-BubR1 wt or the loop mutant showing that the B1-loop enhances KT localization. Scale bar: 10 µm. (**D**) Quantification of BubR1 KT levels in cells treated as in panel **C**. The graph shows mean intensity from three independent experiments. Error bar represents SEM. Values for BubR1 wt are set to 1. (**E**) Representative images of HeLa cells transfected with the indicated GFP-BubR1 constructs, showing that BubR1 B1-loop is independent of Bub1 for its KT localization. In brief, after transfection, cells were depleted of endogenous Bub1 by RNAi, synchronized with a double thymidine block and arrested in mitosis with nocodazole. Scale bar: 10 µm. (**F**) Quantification of BubR1 KT levels in cells treated as in panel **E**. The graph shows mean intensity from three independent experiments. Error bars represent SEM. Values for BubR1$^{wt}$ in non-depleted cells are set to 1. (**G**) Representative images of HeLa cells transfected with GFP-BubR1 B1-loop treated as in panel **E** in the presence (+) or absence (−) of the Mps1 inhibitor Reversine, showing that BubR1 B1-loop KT localization is dependent on Mps1. Scale bar: 10 µm. (**H**) Quantification of BubR1 KT levels in cells treated as in panel **G**. The graph shows mean intensity from two independent experiments. Error bars represent SEM. Values for BubR1$^{wt}$ in non-depleted cells without Reversine (images are shown in *Figure 3—figure supplement 1*, panel **A**) are set to 1. (**I**) Mean duration of mitosis of Flp-In T-REx stable cell lines expressing GFP-BubR1 wt or the loop mutant in the absence of endogenous BubR1 and in the presence of 50 nM nocodazole. Cell morphology was used to measure entry into and exit from mitosis by time-lapse-microscopy (n > 58 per cell line) from three independent experiments. Error bars depict SEM. (**J**) Western Blot of immunoprecipitates (IP) from mitotic Flp-In T-REx cell lines expressing the indicated GFP-BubR1 constructs showing the influence of the loop on the ability to pull down MCC and APC/C components. Tubulin was used as loading control. (**K**) Western Blot of immunoprecipitates (IP) of the APC/C subunit Cdc27 from mitotic Flp-In T-REx cell lines expressing the indicated GFP-BubR1 constructs showing the influence of the loop on the incorporation into APC/C-bound MCC. Tubulin was used as loading control.

The following figure supplement is available for figure 3:

**Figure supplement 1**. Additional characterization of 'loop swap' mutants.

tetratrico peptide repeats (TPRs) near the N-terminus of Bub1 and BubR1 (*Figure 1A*) had been initially identified as primary determinants of kinetochore recruitment (*Kiyomitsu et al., 2007*; *Bolanos-Garcia et al., 2011*), but later shown to be dispensable (*Krenn et al., 2012*, *2014*), a result confirmed here. In our IP experiments, however, we observe that Bub1 interacts with the outer kinetochore more strongly when the TPRs are present, in agreement with our previous studies (*Krenn et al., 2012*, *2014*). The Bub1 TPR region interacts with a short sequence motif of Knl1 named KI1 motif (*Kiyomitsu et al., 2007*; *Krenn et al., 2012*, *2014*). This interaction, whose precise significance is unclear, enhances the SAC response (*Krenn et al., 2014*). Additional evidence of a modest additional role of the TPR in the interaction of Bub1 and BubR1 with kinetochores is presented in *Figure 4—figure supplement 2* and *Figure 4—figure supplement 3*.

Next, we asked if Bub1 binds directly to BubR1. Bub1$^{1–409}$ and BubR1$^{1–571}$, both of which target kinetochores (*Taylor et al., 1998*; *Vanoosthuyse et al., 2004*; *Klebig et al., 2009*; *Malureanu et al., 2009*; *Elowe et al., 2010*) were individually co-expressed with Bub3 in insect cells and purified to homogeneity. In size-exclusion chromatography (SEC) experiments, in which the elution volume reflects macromolecular mass and shape, BubR1$^{1–571}$/Bub3 bound stoichiometrically to Bub1$^{1–409}$/Bub3 (*Figure 4G*), thus demonstrating a direct interaction in the absence of other proteins. In these experiments in vitro, BubR1$^{1–571}$ did not require Bub3 to bind Bub1$^{1–409}$/Bub3 (*Figure 4H*), and therefore in the following SEC experiments, we used BubR1 and BubR1/Bub3 interchangeably (whereas Bub1 was poorly expressed and largely insoluble in insect cells without Bub3). Although this result may suggest that Bub3 is not required for the interaction of BubR1 with Bub1/Bub3, we show and discuss in the context of Figure 6 that a functional B3BD is required for the interaction of BubR1 with Bub1 in living cells.

In agreement with the ability of Bub1$^{209–409}$ to recruit BubR1 to kinetochores, Bub1$^{209–409}$/Bub3 formed a stoichiometric complex with BubR1$^{1–571}$/Bub3 (*Figure 4I*) or BubR1$^{1–571}$ (*Figure 4—figure supplement 1*, panel A), further showing that the Bub1 TPR region is dispensable. Neither the isolated B3BD of Bub1 (Bub1$^{209–270}$/Bub3, *Figure 4—figure supplement 1B*) nor the isolated CTE (Bub1$^{271–409}$, *Figure 4J*) bound BubR1$^{1–571}$/Bub3, indicating that both regions contribute to BubR1 binding in vitro. Additional SEC experiments supporting this conclusion are shown in *Figure 4—figure supplement 1*, panels C–D.

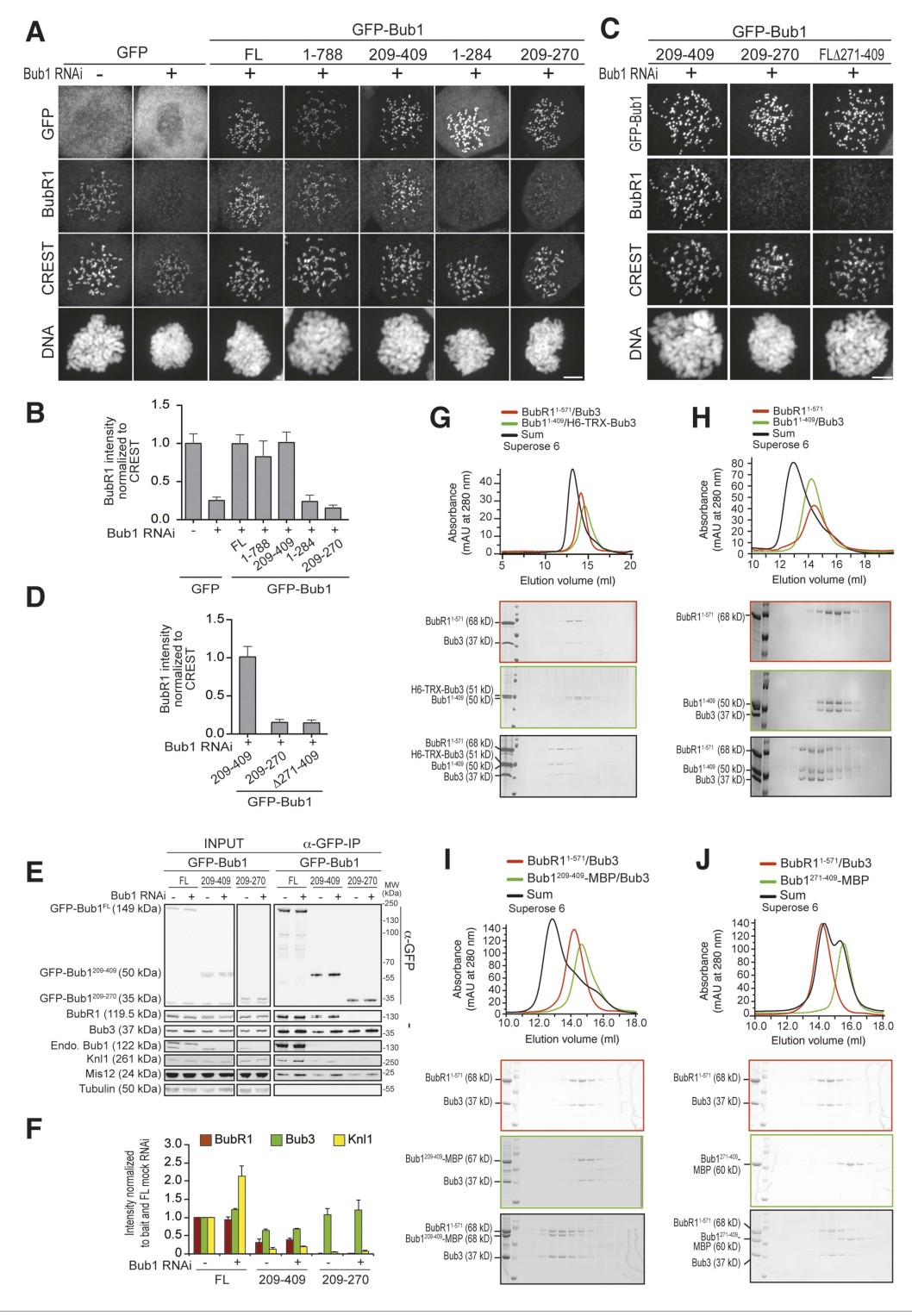

**Figure 4**. A minimal BubR1-binding region of Bub1. (**A** and **C**) Representative images of stable Flp-In T-REx cell lines expressing the indicated GFP-Bub1 constructs after treatment with nocodazole, showing that Bub1$^{209-409}$ is sufficient to recruit BubR1 (panel **A**) and that residues 271–409 are essential for this function (panel **C**). Scale bar: 10 μm. (**B** and **D**) Quantification of BubR1 KT levels in cells treated as in panels **B** and **D**, respectively. The graphs show mean intensity of two independent experiments, the error bars indicate SEM. The mean value for non-depleted cells expressing GFP (panel **B**) or GFP-Bub1$^{209-409}$ (panel **D**) is set to 1. (**E**) Western blot of immunoprecipitates (IP) from

*Figure 4. continued on next page*

*Figure 4. Continued*

mitotic Flp-In T-REx cell lysates expressing the indicated GFP-Bub1 constructs in the presence or absence of endogenous Bub1, showing that Bub1$^{209-409}$ is sufficient to pull down BubR1. Tubulin was used as loading control. (**F**) Quantification of the Western blot in panel **E**. The amounts of co-precipitating BubR1, Bub3, and Knl1 were normalized to the amount of GFP-Bub1 bait present in the IP. Values for GFP-Bub1 FL in non-depleted cells are set to 1. The graph shows mean intensity of two independent experiments. Error bars represent SD. (**G**) BubR1$^{1-571}$/Bub3 and Bub1$^{1-409}$/Bub3 interact in size exclusion chromatography, which separates proteins based on size and shape. H6 and TRX are tags used for protein purification and expression. (**H**) BubR1$^{1-571}$ and Bub1$^{1-409}$/Bub3 interact in size exclusion chromatography. (**I**) BubR1$^{1-571}$/Bub3 and Bub1$^{209-409}$/Bub3 interact in size exclusion chromatography. (**J**) BubR1$^{1-571}$/Bub3 and Bub1$^{271-409}$ do not interact in size exclusion chromatography. MBP—maltose binding protein, mAu—milliabsorbance unit.

The following figure supplements are available for figure 4:

**Figure supplement 1**. Additional chromatographic experiments.

**Figure supplement 2**. The TPR domain of Bub1 influences KT binding affinity in addition to the loop region.

**Figure supplement 3**. The TPR domain of BubR1 influences kinetochore binding affinity in addition to the loop region.

## A minimal Bub1-binding region of BubR1

To identify a minimal kinetochore-targeting region of BubR1, we created GFP fusions of several deletion mutants of BubR1 (*Figure 1—figure supplement 1C–D*) and tested their localization to kinetochores. BubR1$^{362-571}$, which contains the B3BD and a ~140-residue CTE, localized to kinetochores in a Bub1-dependent manner (*Figure 5A–B*). Shorter fragments of BubR1, including the isolated B3BD (BubR1$^{362-431}$, see also *Figure 1J*) and the isolated CTE (BubR1$^{432-571}$), did not localize to kinetochores (*Figure 5B–C*). In agreement with the kinetochore recruitment assay, SEC experiments showed binding of BubR1$^{362-571}$ but not of the isolated B3BD (BubR1$^{362-431}$/Bub3) or the isolated CTE (BubR1$^{432-571}$) to Bub1$^{1-409}$/Bub3 (*Figure 5D–F*).

## A pseudo-symmetric Bub1–BubR1 interaction

Our results predict that BubR1$^{362-571}$ and Bub1$^{209-409}$ ought to be sufficient for the Bub1/BubR1 interaction in vitro. Indeed, BubR1$^{362-571}$ and Bub1$^{209-409}$/Bub3 interacted stoichiometrically in SEC runs (*Figure 6A*). A summary of the properties of the crucial Bub1 and BubR1 segments discussed in the last two sections is presented in *Figure 6B*. An alignment of the interacting domains of Bub1 and BubR1 (residues 209–409 and 362–571, respectively; the alignment was obtained with programs Muscle [*Edgar, 2004*] and JPRED [*Cole et al., 2008*]) shows that their sequences are structurally equivalent (*Figure 6—figure supplement 1A*). Both start with the B3BD, continue with a segment predicted to adopt a helical conformation and end with a region predicted to lack defined secondary structure. We surmise that the ability of modern-day Bub1 and BubR1 to form heterodimers using these structurally equivalent ('pseudo-symmetric') segments may reflect the ability of their ancestor to form homodimers, similarly to what is observed in cohesins.

## Functional dissection of kinetochore recruitment of BubR1

Deletion of residues 432–484 of BubR1 (GFP-BubR1$^{\Delta432-484}$) in the predicted helical region impaired kinetochore localization of BubR1 (*Figure 6C–D*). Additionally, mutations in the B3BD of BubR1 (E409K + E413K) known to impair Bub3 binding (*Taylor et al., 1998*; *Larsen et al., 2007*) prevented kinetochore localization of full-length BubR1 (*Figure 6E–F*) and of BubR1$^{362-571}$ (*Figure 6—figure supplement 1*, panels B–C). Thus, Bub3 binding is necessary for efficient kinetochore localization of BubR1 even if it may appear to be dispensable for the interaction in vitro (*Figure 4H*). We note that in those SEC binding experiments in vitro in which we used isolated BubR1 rather than BubR1/Bub3, Bub3 might have exchanged from the Bub1/Bub3 complex to reconstitute BubR1/Bub3. Alternatively, the relatively high protein concentrations in the SEC experiments (5–15 μM) may effectively compensate for reduced binding affinity when BubR1 is devoid of Bub3.

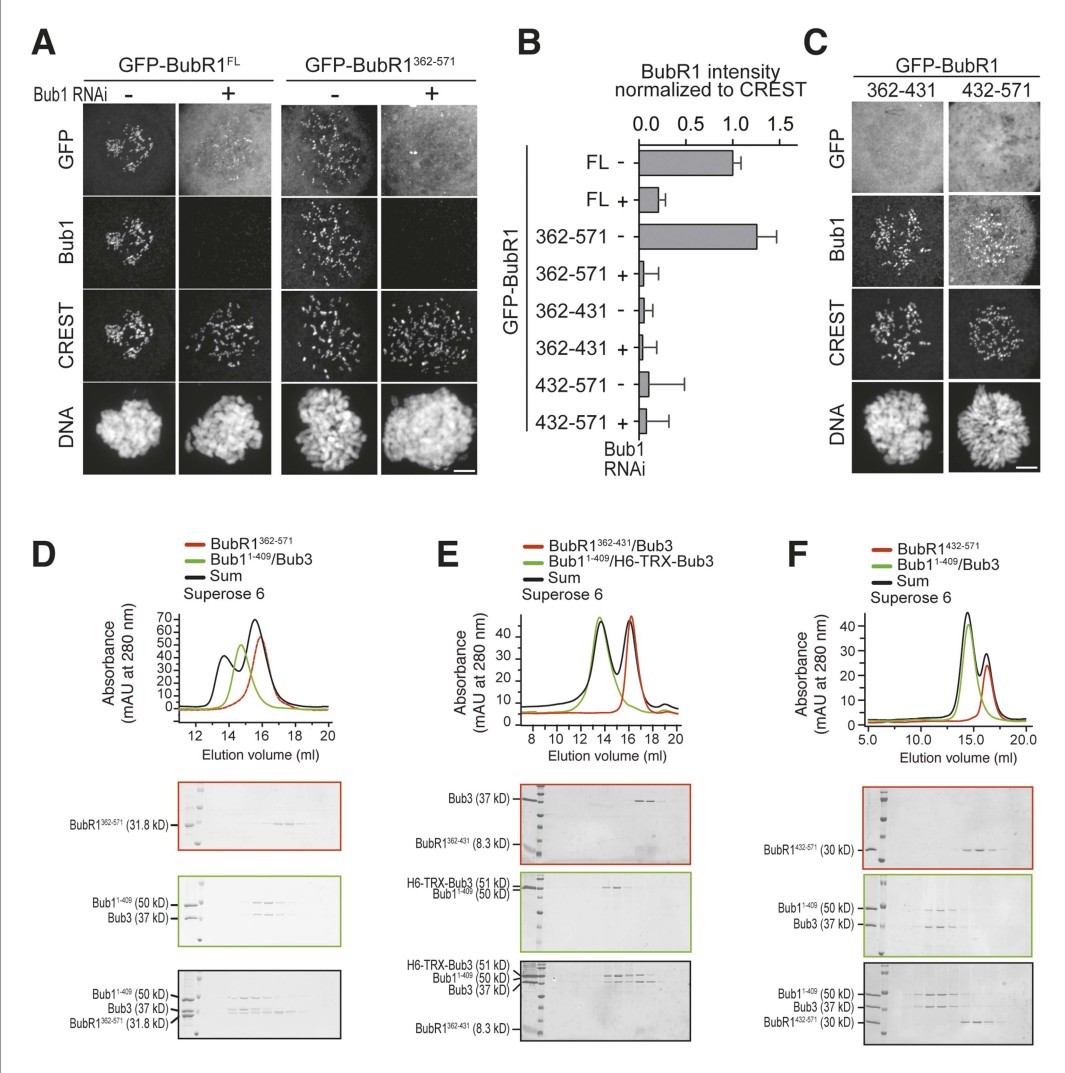

**Figure 5**. A minimal Bub1-binding region of BubR1. (**A** and **C**) Representative images of HeLa cells transfected with the indicated GFP-BubR1 constructs. Cells were treated as described in *Figure 3E*. BubR1$^{362–571}$ is the minimal construct that is able to localize to KTs in the presence of Bub1. Scale bar: 10 μm. (**B**) Quantification of BubR1 KT levels in cells treated as in panels **A** and **C**. The graph shows mean intensity of at least two independent experiments, error bars depict SEM. Values for GFP-BubR1 FL in non-depleted cells are set to 1. (**D**) BubR1$^{362–571}$ and Bub1$^{1–409}$/Bub3 interact in size exclusion chromatography. (**E**) BubR1$^{362–431}$/Bub3 and Bub1$^{1–409}$/Bub3 do not interact in size exclusion chromatography. (**F**) BubR1$^{432–571}$ and Bub1$^{1–409}$/Bub3 do not interact in size exclusion chromatography. mAU—milliabsorbance unit.

Regardless of the precise explanation, BubR1 clearly requires Bub3 for efficient kinetochore localization. This requirement may reflect a direct contribution to the interaction with Bub1 or alternatively a residual ability of BubR1/Bub3 to bind phosphorylated motifs on Knl1 or other kinetochore proteins. To try distinguishing between these possibilities, we targeted a Lac repressor (LacI) fusion of Bub1 to an ectopic Lac operator (LacO) site on the chromosome distinct from centromeres. LacI-Bub1 recruited BubR1 to the ectopic site (*Figure 6G–H*, quantified in *Figure 6—figure supplement 1D*), thus suggesting that kinetochores are not required for the Bub1/BubR1 interaction, in agreement with a previous study in *Schizosaccharomyces pombe* (*Rischitor et al., 2006*). This result suggests that Bub1/Bub3 is sufficient for recruitment of BubR1/Bub3, and that the role of Bub3 in recruitment of BubR1 might be direct and not through Knl1. Mutation of the Bub3-binding site of the LacI-Bub1 fusion (Bub1$^{E252K}$) prevented BubR1 recruitment, suggesting

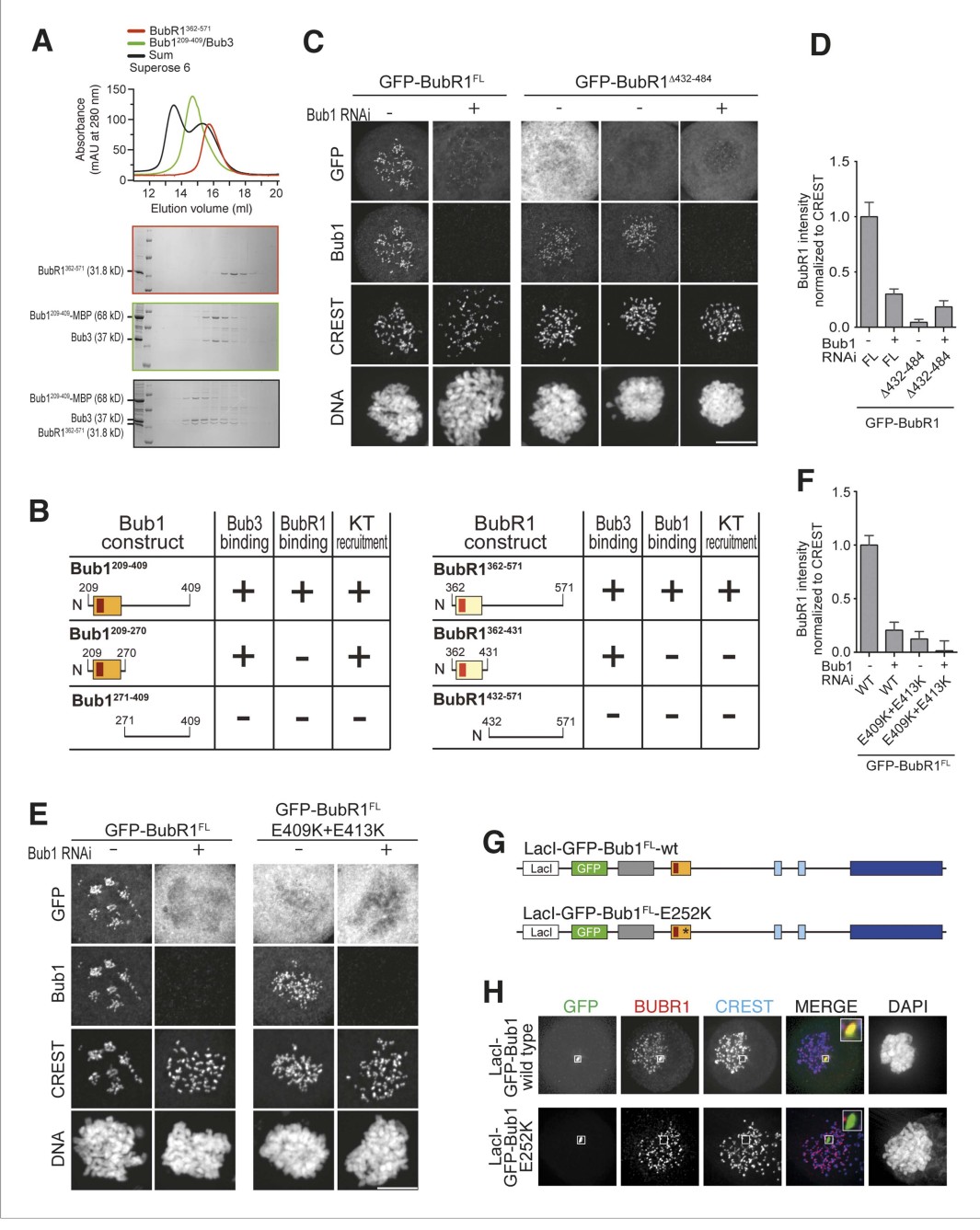

Figure 6. A pseudo-symmetric Bub1–BubR1 interaction. (A) The identified minimal constructs BubR1$^{362–571}$ and Bub1$^{209–409}$/Bub3 interact in size exclusion chromatography. (B) Summary of the behavior of the indicated Bub1 and BubR1 constructs. (C and E) Representative images of HeLa cells transfected with the indicated GFP-BubR1 constructs showing that neither BubR1$^{Δ432–484}$ (panel C), which lacks the predicted helical segment of the C-terminal extension, nor BubR1$^{E409K+E413K}$ (panel E), which is not able to bind Bub3, are able to localize to KTs. Cells were treated as in Figure 3E. For BubR1$^{Δ432–484}$ two different expression levels are depicted in the non-depleted condition. Scale bar: 10 μm. (D and F) Quantification of BubR1 KT levels in cells treated as in panels C and E, respectively. The graph shows mean intensity from at least two independent experiments. Error bars represent SEM. Values for BubR1$^{FL}$ in non-depleted cells are set to 1. (G) Domain organization of LacI-GFP-Bub1 constructs. (H) LacI-Bub1$^{wt}$ recruits BubR1 to the Lac-Operator, whereas Bub1$^{E252K}$, which cannot bind Bub3, does not.

The following figure supplement is available for figure 6:

Figure supplement 1. Alignment of the Bub1 and BubR1 interaction domains.

that Bub3 is required for a robust interaction between Bub1 and BubR1 on both sides of the complex (*Figure 6G–H*).

BubR1 is important for the SAC and for establishing bi-orientation. We therefore asked if perturbing its kinetochore localization impaired these processes. HeLa cells depleted of endogenous BubR1 failed to arrest in mitosis in the presence of nocodazole (*Figure 7A*). SAC proficiency was re-established upon expression of GFP-BubR1 but not GFP-BubR1$^{E409K+E413K}$. GFP-BubR1$^{\Delta432–484}$, on the other hand, restored a robust SAC response. IP experiments indicated that both GFP-BubR1$^{E409K+E413K}$ and GFP-BubR1$^{\Delta432–484}$ were unable to interact with Bub1 or Knl1, as expected based on their localization behavior. However, GFP-BubR1$^{E409K+E413K}$ was also significantly impaired in its ability to interact with two additional MCC subunits (Mad2 and Cdc20) and was even more dramatically impaired in its ability to interact with the APC/C. Conversely, GFP-BubR1$^{\Delta432–484}$ interacted with the MCC subunits (including Bub3, as expected) and with the APC/C at levels that were comparable to those of the wild-type GFP-BubR1 control (*Figure 7B*). Thus, two distinct mutants of BubR1 both impaired in kinetochore localization have uncorrelated behaviors with regard to SAC proficiency. On the other hand, because GFP-BubR1$^{E409K+E413K}$ does not bind Bub3, while GFP-BubR1$^{\Delta432–484}$ does (*Figure 7B*, quantified in *Figure 7C*), it appears that at least in human cells BubR1 needs to bind Bub3 to become incorporated into the MCC.

In addition to playing a role in the SAC, BubR1 is important for establishing bi-orientation in HeLa cells, and this role requires its kinetochore recruitment (*Johnson et al., 2004*; *Lampson and Kapoor, 2005*; *Meraldi and Sorger, 2005*; *Suijkerbuijk et al., 2012b*; *Kruse et al., 2013*; *Xu et al., 2013*). Both GFP-BubR1$^{E409K+E413K}$ and GFP-BubR1$^{\Delta432–484}$, neither of which localizes to kinetochores, failed to complement the deficits in promoting the formation of stable kinetochore–microtubule attachments observed in cells depleted of endogenous BubR1 (*Figure 7D*). The role of BubR1 in kinetochore–microtubule attachment has been attributed to its interaction with the B56 regulatory subunit of a complex of protein phosphatase 2A (PP2A$^{B56}$) (*Suijkerbuijk et al., 2012b*; *Kruse et al., 2013*; *Xu et al., 2013*; *Espert et al., 2014*; *Nijenhuis et al., 2014*). Recently, this function of BubR1 has been further implicated in SAC silencing (*Espert et al., 2014*; *Nijenhuis, et al., 2014*). We asked if the defect of the BubR1$^{\Delta432–484}$ mutant in supporting kinetochore–microtubule attachment correlated with a defective interaction with PP2A$^{B56}$. Indeed, the levels of the B56 regulatory subunit in IPs of the GFP-BubR1$^{\Delta432–484}$ mutant were much lower than those of the wild-type protein (*Figure 7E*).

## Discussion

New genes are frequently created by duplication (*Conant and Wolfe, 2008*). After duplication, the paralogs (i.e., the genes generated by the duplication event) may diverge and sub-functionalize, thus allowing specialization in a subset of the functions originally provided by the singleton (the single ancestor gene before duplication) (*Conant and Wolfe, 2008*). The resolution of adaptive conflicts between different functions of the singleton has been identified as a beneficial consequence of gene duplication (*Hittinger and Carroll, 2007*). Detailed molecular illustrations of this process, however, are rare.

Our analysis of the mechanism of recruitment of Bub1 and BubR1 illustrates how the duplication of an ancestor gene and subsequent divergence was exploited during evolution to create two gene products with highly diversified and efficient functions, including monitoring checkpoint status through stable recruitment to unattached kinetochores (Bub1), effecting APC/C inhibition through incorporation into the MCC (BubR1), stabilization of microtubule attachment (Bub1 and BubR1, possibly through different mechanisms), and SAC silencing (BubR1). The occurrence of nine distinct duplications of the Bub1 and BubR1 ancestor (*Suijkerbuijk et al., 2012a*) likely reflects an extreme evolutionary pressure to separate functions that were originally condensed in a single gene. Investigating to what extent sub-functionalization after each of the nine distinct duplication events of the Bub1 and BubR1 ancestor followed similar or divergent paths will be an interesting direction for future studies.

Previously, the loss of kinase activity specifically in BubR1 but not Bub1 was identified as a manifestation of the divergence of these proteins (*Vleugel et al., 2012*; *Suijkerbuijk et al., 2012b*). Here, we have considerably extended our understanding of this divergence by showing that the loop motifs in the B3BDs of Bub1 and BubR1 modulate the interaction of Bub3 with MELT$^P$. A suboptimal loop region for MELT$^P$ binding in BubR1 makes it depend on an alternative mechanism for kinetochore localization. This alternative is the direct interaction with Bub1, extensively characterized

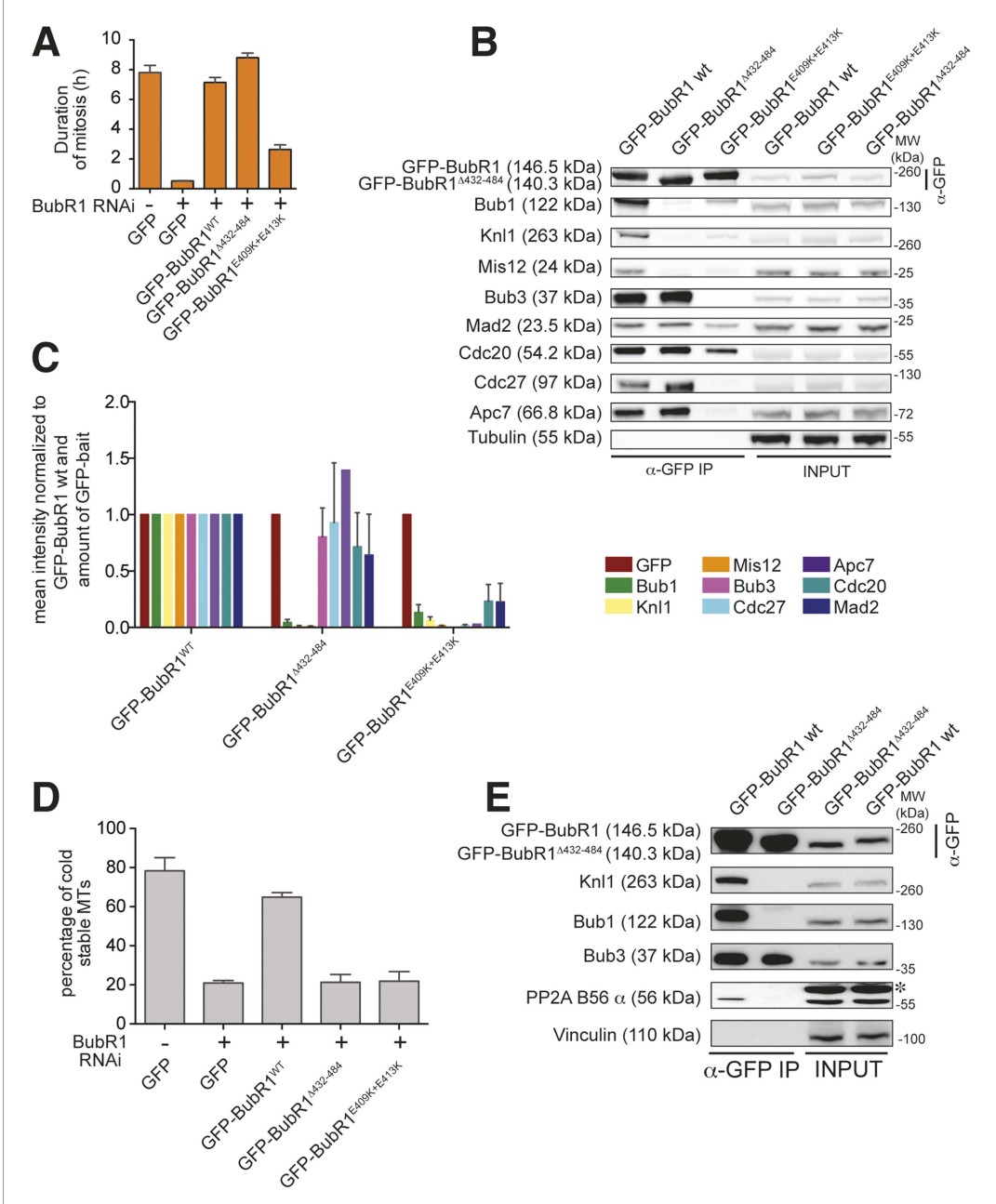

**Figure 7**. Functional characterization of BubR1 mutants. (**A**) Mean duration of mitosis of Flp-In T-REx stable cell lines expressing GFP-BubR1 wt or the indicated mutants in the absence of endogenous BubR1 and in the presence of 50 nM nocodazole. Cell morphology was used to measure entry into and exit from mitosis by time-lapse-microscopy (n > 44 per cell line per experiment) from at least three independent experiments. Error bars depict SEM.
(**B**) Western blot of immunoprecipitates (IP) from mitotic Flp-In T-REx cell lines expressing the indicated GFP-BubR1 constructs. Tubulin was used as loading control. (**C**) Quantification of the Western Blot in *Figure 7B*. The amounts of co-precipitating proteins were normalized to the amount of GFP-BubR1 bait present in the IP. Values for GFP-BubR1 wt are set to 1. The graphs show mean intensity of two independent experiments. Error bars represent SEM.
(**D**) Analysis of cold-stable microtubules in cells expressing the indicated GFP-BubR1 constructs. (**E**) Western Blot of immunoprecipitates (IP) from mitotic Flp-In T-REx cell lines expressing the indicated GFP-BubR1 constructs. The asterisk represents an unspecific band recognized by the PP2A antibody.

in *Figures 4–6*. By swapping loop motifs, we created a gain-of-function mutant of BubR1 that can bind kinetochores in the absence of Bub1, and a loss-of-function mutant of Bub1 severely impaired in autonomous kinetochore binding. The failure of the BubR1 gain-of-function mutant to complement the depletion of wild-type BubR1 strongly suggests that the evolutionary divergence of Bub1 and BubR1, with its specific effects on kinetochore recruitment, is functionally relevant.

The mechanism of BubR1 localization depends on a direct, pseudo-symmetric hetero-dimerization interaction with Bub1 (*Figure 8A*). It involves equivalent segments of Bub1 and BubR1 comprising their B3BD/GLEBS motif and a CTE whose first part is predicted to be helical. The interaction of Bub1 and BubR1 requires that each has a bound Bub3. Thus, Bub3 has at least three distinct functions: (1) it recruits Bub1 to MELT$^P$ motifs of Knl1; (2) it contributes to the dimerization of Bub1 with BubR1 required for kinetochore recruitment of the latter; and (3) in complex with BubR1, it has an additional hitherto unknown function in the SAC. This function was exposed by the behavior of the BubR1$^{E409K+E413K}$ mutant. The SAC defect observed with this mutant is unlikely to be a consequence of impaired kinetochore localization, because another kinetochore-localization impaired mutant, BubR1$^{Δ432–484}$, was checkpoint proficient. We surmise that BubR1-bound Bub3 is involved in an unknown aspect of the SAC mechanism downstream of kinetochores, possibly having to do with MCC formation or APC/C binding, as recently proposed (*Han et al., 2014*). In analogy with the role of the Bub1 loop motif in modulating the function of Bub3 as a MELT$^P$ receptor, we speculate that the BubR1 loop motif influences the specificity of BubR1-bound Bub3 for additional SAC-relevant targets. The fact that the 'loop swap' BubR1 mutant is unable to sustain the SAC provides evidence for this hypothesis. The properties of the loop region in the recently discovered protein BuGZ, which contains a B3BD/GLEBS motif that interacts with Bub3, are also of interest (*Jiang et al., 2014*; *Toledo et al., 2014*).

Recruitment of PP2A$^{B56}$ by BubR1 has been recently implicated in SAC silencing through a mechanism ultimately impinging on dephosphorylation of the MELT$^P$ motifs (*Espert et al., 2014*; *Nijenhuis et al., 2014*). The apparent checkpoint proficiency of BubR1$^{Δ432–484}$ may indicate that BubR1 localization to kinetochores is not essential for checkpoint function, as it implies that substantial amounts of MCC can be generated even when BubR1/Bub3 cannot be recruited to kinetochores. However, we note that this mutant interacts only weakly with PP2A$^{B56}$ and might therefore carry an additional checkpoint-silencing defect obscuring an underlying SAC defect. Furthermore, we cannot exclude that residues in the deleted segments (432–484) of BubR1 are normally involved in an intra-molecular control mechanism that couples the activation of wild-type BubR1 to kinetochore recruitment.

Whether or not BubR1 kinetochore recruitment is important for SAC function, it is clearly essential for kinetochore–microtubule attachment and bi-orientation. Previously, it has been shown that BubR1 promotes bi-orientation through recruitment of PP2A$^{B56}$ (*Suijkerbuijk et al., 2012b*; *Kruse et al., 2013*; *Xu et al., 2013*), which counteracts Aurora B activity and thus stabilizes kinetochore–microtubule attachments (*Lampson and Kapoor, 2005*; *Foley et al., 2011*). Our results demonstrate that BubR1$^{Δ432–484}$, which cannot interact with Bub1 and localize to kinetochores, has a strong defect in kinetochore–microtubule attachment that correlates with a defective interaction with PP2A$^{B56}$.

Dissecting the requirement for kinetochore recruitment of the SAC subunits is instrumental for distinguishing their roles in the SAC from their roles in chromosome bi-orientation (*Brady and Hardwick, 2000*; *De Antoni et al., 2005*; *London et al., 2012*; *Shepperd et al., 2012*; *Yamagishi et al., 2012*; *Nijenhuis et al., 2013*; *London and Biggins, 2014*; *Moyle et al., 2014*). By describing the mechanism of BubR1 recruitment and the role of the loop motif of Bub1 and BubR1 in modulating the affinity for kinetochores, this study fills an important gap. We and others have previously shown that two conformers of Mad2, O-Mad2 and C-Mad2, form an asymmetric conformational dimer (*Luo et al., 2004*; *De Antoni et al., 2005*). In this reaction, Mad1 acts as a stable placeholder for C-Mad2 (*De Antoni et al., 2005*). Once at kinetochores, the Mad1/C-Mad2 complex recruits a high-turnover cytosolic form of O-Mad2 and converts it into the active C-Mad2 form, which targets Cdc20, thus overcoming a rate-limiting step towards the formation of MCC (*De Antoni et al., 2005*; *Mapelli et al., 2007*; *Simonetta et al., 2009*). This scheme, summarized in *Figure 8B*, identifies Mad1/Mad2 as a 'template' for the establishment of the Cdc20/C-Mad2 complex, a structural 'copy' of the Mad1/Mad2 complex (*De Antoni et al., 2005*; *Musacchio and Salmon, 2007*).

Even if directed primarily towards a different function (SAC signaling rather than chromosome bi-orientation), we note that this pattern is remarkably similar to that emerging from the mechanism of BubR1 recruitment by Bub1, in which a stable Bub1/Bub3 complex at kinetochores recruits a rapidly

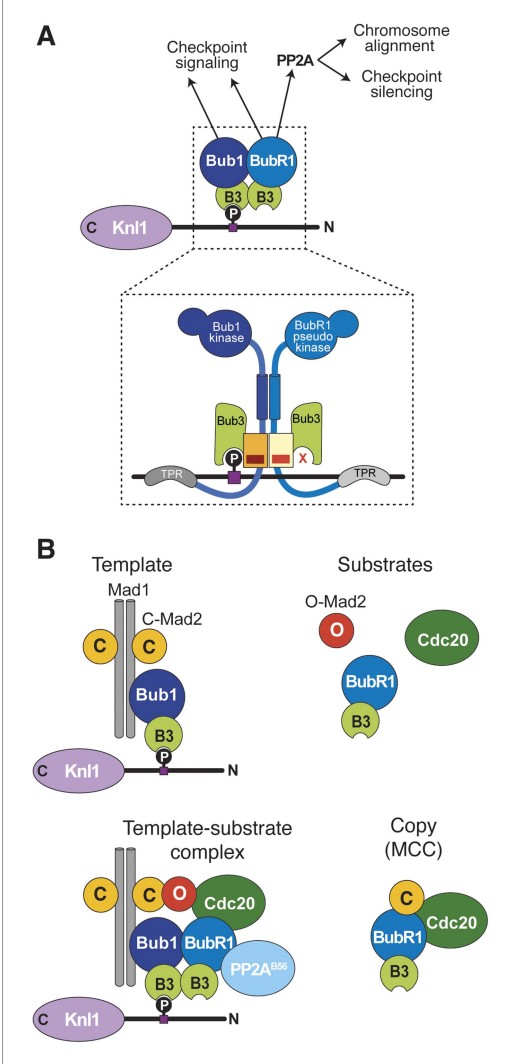

**Figure 8**. Extension of the template model. (**A**) Model of the Bub1–BubR1 interaction. The upper part shows the described KT recruitment mechanism of BubR1/Bub3, which in turn recruits the phosphatase PP2A, to a Bub1/Bub3 complex on Knl1. The lower part depicts a close-up of the identified pseudo-symmetric Bub1–BubR1 interaction, which involves equivalent segments of Bub1 and BubR1 comprising the Bub3 binding domain and a C-terminal extension whose first part is predicted to have a helical fold in both proteins. The presence of Bub3 on both proteins seems to be essential for this interaction, although due to different reasons (for more explanations see text). The TPR regions of human Bub1 and BubR1 bind to non-conserved short motifs of Knl1 named KI1 and KI2, respectively (*Kiyomitsu et al., 2007*; *Krenn et al., 2012, 2014*). (**B**) Extension of the template model. Mad1/C-Mad2 at KTs is known to act as a template for the establishment of the Cdc20/C–Mad2 interaction. This seems similar to the BubR1 recruitment mechanism, wherein Bub1/Bub3 recruits BubR1/Bub3 through a pseudo-symmetric

cycling BubR1/Bub3 complex. The MCC, made of Cdc20/C-Mad2 and BubR1/Bub3, can be interpreted as a 'copy' of kinetochore-bound 'templates' made of Mad1/C-Mad2 and Bub1/Bub3 complexes. Whether such templates engage in a complex at kinetochores is unclear but plausible. Even if we did not identify Mad1 in our mass spectrometry analysis of Bub1, we recently identified both Mad1 and Bub1 in precipitates of the N-terminal segment of Knl1 (*Krenn et al., 2014*). Furthermore, Mad1 and Bub1 have been shown to interact directly in *S. cerevisiae* and *Caenorhabditis elegans* (*Brady and Hardwick, 2000*; *London et al., 2012*; *Moyle et al., 2014*). Our future studies will aim to investigate the significance of the copy–template molecular relationship for SAC signaling and chromosome bi-orientation.

## Materials and methods

### Mammalian plasmids

Plasmids were derived from the pCDNA5/FRT/TO-EGFP-IRES, a previously modified version (*Krenn et al., 2012*) of the pCDNA5/FRT/TO vector (Invitrogen, Carlsbad, CA). To create N-terminally-tagged EGFP Bub1 and BubR1 truncation constructs, Bub1 and BubR1 sequences were obtained by PCR amplification from the previously generated pCDNA5/FRT/TO-EGFP-Bub1-IRES and pCDNA5/FRT/TO-EGFP-BubR1-IRES vector, respectively (*Krenn et al., 2012*) and subcloned in frame with the GFP-tag. Mutations and deletions within the Bub1 and BubR1 constructs were generated by standard site-directed mutagenesis or by a mutagenesis protocol (*Liu and Naismith, 2008*). All Bub1 constructs were RNAi resistant (*Kiyomitsu et al., 2007*). BubR1-expressing constructs were made siRNA-resistant by changing the sequence targeted by the RNAi oligos to 'AACGTGCCTTCGAGTACGAGA'. pCDNA5/FRT/TO-based plasmids were used for generation of stable cell lines, as well as for transient transfection. All plasmids were verified by sequencing.

### Cell culture and transfection

HeLa cells were grown in DMEM (PAN Biotech, Aidenbach, Germany) supplemented with 10% FBS (Clontech, part of Takara Bio group, Shiga, Japan), penicillin and streptomycin (GIBCO, Carlsbad, CA), and 2 mM L-glutamine (PAN Biotech). For all plasmid transfections of HeLa cells, X-tremeGENE transfection reagent (Roche, Basel, Switzerland) was used at a 3:1 ratio with plasmid DNA. Flp-In T-REx HeLa cells used to generate stable doxycycline-inducible cell lines were a gift from

*Figure 8. Continued*

interaction. Ultimately, the entire MCC (BubR1/Bub3 and Cdc20/C-Mad2) may represent the copy of a KT template consisting of Bub1/Bub3 and Mad1/C-Mad2.

SS Taylor (University of Manchester, Manchester, England, UK). Flp-In T-REx host cell lines were maintained in DMEM with 10% tetracycline-free FBS (Clontech) supplemented with 50 µg/ml Zeocin (Invitrogen). Flp-In T-REx HeLa expression cell lines were generated as previously described (*Krenn et al., 2012*). Gene expression was induced by addition of 0.2–0.5 µg/ml doxycycline (Sigma, St. Louis, MO) for 24 hr siBUB1 (Dharmacon, part of GE Healthcare, Piscataway, NJ; 5'-GGUUGCCAACACAAGUUCU-3') or siBUBR1 (Dharmacon; 5'-CGGGCAUUUGAAUAUGAAA-3') duplexes were transfected with Lipofectamine 2000 (Invitrogen) at 50 nM for 24 hr.

For experiments in HeLa cells, cells were synchronized with a double thymidine arrest 5 hr after transfection with siRNA duplexes. In brief, after washing the cells with PBS they were treated with thymidine for 16 hr and then released into fresh medium. 3 hr after the release, 50 nM siRNA duplexes were transfected again. 5 hr after transfection, cells were treated with thymidine for 16 hr and afterwards released in fresh medium. Unless differently specified, nocodazole (Sigma–Aldrich) was used at 3.3 µM. MG132 (used at 5–10 µM) was obtained from Calbiochem, thymidine (2 mM) was purchased from Sigma–Aldrich. Reversine (Calbiochem, part of EMD Biosciences, Darmstadt, Germany) was used at 0.5 µM.

## Immunoprecipitation and immunoblotting

To generate mitotic populations for immunoprecipitation experiments, cells were treated with 330 nM nocodazole for 16 hr. Mitotic cells were then harvested by shake off and lysed in lysis buffer (150 mM KCl, 75 mM Hepes, pH 7.5, 1.5 mM EGTA, 1.5 mM $MgCl_2$, 10% glycerol, and 0.075% NP-40 supplemented with protease inhibitor cocktail [Serva, Heidelberg, Germany] and PhosSTOP phosphatase inhibitors [Roche]). Extracts were precleared using a mixture of protein A–Sepharose (CL-4B; GE Healthcare) and protein G-Sepharose (rec-Protein G-Sepharose 4B; Invitrogen) for 1 hr at 4°C. Subsequently, extracts were incubated with GFP-Traps (ChromoTek, Martinsried, Germany; 3 µl/mg of extract) for 3 hr at 4°C. Immunoprecipitates were washed with lysis buffer and resuspended in sample buffer, boiled and analyzed by SDS-PAGE and Western blotting using 4–12% gradient gels (NuPAGE Bis-Tris Gels, Life technologies, Carlsbad, CA). For Cdc27 IPs cells were synchronized by addition of the CDK1-inhibitor RO3306 (Calbiochem) for 15 hr and subsequently released into 330 nM nocodazole for 2–3 hr before harvesting by shake off. Cells were lysed in lysis buffer (described above), and extracts were precleared with protein G-Sepharose for 1 hr at 4°C. Afterwards, extracts were incubated with 1.5 µg/mg of the Cdc27 primary antibody (mouse monoclonal, BD) for 2 hr at 4°C. Subsequently, protein G-Sepharose was added for 4 hr at 4°C. Immunoprecipitates were washed and analyzed as described above. The following antibodies were used: anti-GFP (in house made rabbit polyclonal antibody; 1:1000–3000), anti-Mis12 (in house made mouse monoclonal antibody; clone QA21-74-4-3; 1:1000), anti-Knl1-N (in house made rabbit polyclonal SI0787 antibody; 1:1000), anti-Bub1 (rabbit polyclonal; Abcam, Cambridge, UK; 1:5000), anti-BubR1 (mouse monoclonal; BD; 1:1000), anti-Bub3 (mouse monoclonal; BD; 1:1000), anti-Tubulin (mouse monoclonal; Sigma; 1:8000), anti-Apc7 (in house made rabbit polyclonal antibody SI0651, 1:500), anti-Cdc20 (mouse monoclonal, Santa Cruz, Dallas, TX, 1:500), anti-Mad2 (in house made mouse monoclonal antibody, clone AS55-A12, 1:500), anti-Cdc27 (mouse monoclonal, BD; 1:1000-3000), anti-PP2A$^{B56\alpha}$ (rabbit polyclonal; Bethyl, Montgomery, TX; 1:1000). Secondary antibodies were anti-mouse (Amersham, part of GE Healthcare) and anti-rabbit (Amersham) affinity purified with horseradish peroxidase conjugate (working dilution 1:10000) or Protein G with horseradish peroxidase conjugate (Life technologies) (working dilution 1:6000). After incubation with ECL Western blotting system (GE Healthcare), images were acquired with ChemiBIS 3.2 (DNR Bio-Imaging Systems, Jerusalem, Israel) in 16-bit TIFF format. Levels of images were adjusted using ImageJ software and then cropped and converted to 8-bit. Unmodified 16-bit TIFF images were used for quantification with ImageJ software. Measurements were graphed with Excel (Microsoft, Seattle, WA) and GraphPad Prism version 6.0 for Mac OS X (GraphPad Software, San Diego California USA).

## Live cell imaging

Cells were plated on a 24-well µ-Plate (Ibidi, Martinsried, Germany). Drugs were diluted in $CO_2$ Independent Medium (Gibco) and added to the cells 1 hr before filming. Cells were imaged every 20 to 30 min in a heated chamber (37°C) on a 3i Marianas system (Intelligent Imaging Innovations Inc.,

Göttingen, Germany) equipped with Axio Observer Z1 microscope (Zeiss), Plan-Apochromat 40×/ 1.4NA oil objective, M27 with DIC III Prism (Zeiss, Oberkochen, Germany), Orca Flash 4.0 sCMOS Camera (Hamamatsu, Hamamatsu City, Japan) and controlled by Slidebook Software 5.5 (Intelligent Imaging Innovations Inc). For cells expressing the GFP-BubR1 proteins, only cells in which kinetochores were visible were considered for the analysis.

## Immunofluorescence

HeLa and Flp-In T-REx HeLa cells were grown on coverslips precoated with poly-D-Lysine (Millipore, 15 µg/ml) and poly-L-Lysine (Sigma), respectively. For the experiments with HeLa cells, cells were synchronized with a double thymidine block and after release from that arrested in prometaphase by the addition of 330 nM nocodazole for 3 hr. For all other experiments, asynchronously growing cells were arrested in prometaphase by the addition of nocodazole for 3–4 hr and fixed using 4% paraformaldehyde. Cells were stained for Bub1 (mouse, ab54893, 1:400), BubR1 (rabbit, Bethyl A300-386A, 1:1000), CREST/anti-centromere antibodies (Antibodies, Inc., Davis, CA, 1:100), diluted in 2% BSA-PBS for 1.5 hr. Goat anti-human and chicken anti-rabbit Alexa Fluor 647 (Invitrogen), goat anti-rabbit and anti-mouse RRX, and donkey anti-human Alexa Fluor 405 (Jackson ImmunoResearch Laboratories, Inc., West Grove, PA) were used as secondary antibodies. DNA was stained with 0.5 µg/ml DAPI (Serva), and coverslips were mounted with Mowiol mounting media (Calbiochem). Cells were imaged at room temperature using a spinning disk confocal device on the 3i Marianas system equipped with an Axio Observer Z1 microscope (Zeiss), a CSU-X1 confocal scanner unit (Yokogawa Electric Corporation, Tokyo, Japan), Plan-Apochromat 63× or 100×/1.4NA Oil Objectives (Zeiss), and Orca Flash 4.0 sCMOS Camera (Hamamatsu). Images were acquired as z-sections at 0.27 µm. Images were converted into maximal intensity projections, exported, and converted into 8-bit. Quantification of kinetochore signals was performed on unmodified 16-bit z-series images using Imaris 7.3.4 32-bit software (Bitplane, Zurich, Switzerland). After background subtraction, all signals were normalized to CREST. At least 138 kinetochores were analyzed per condition. Measurements were exported in Excel (Microsoft) and graphed with GraphPad Prism 6.0 (GraphPad Software, San Diego California USA).

For analysis of cold-stable microtubules, cells that were synchronized with a single Thymidine arrest, released for 6.5 hr and kept for 4 hr in 5 µM MG132, were incubated for 5 min on ice in medium with 10 mM HEPES pH 7.5 and then directly fixed in 4% PFA. Cells were stained for Tubulin (mouse, Sigma T9026, 1:5000) and CREST. DNA was labeled with DAPI. CREST staining was used to identify kinetochores in image z-stacks to count kinetochores attached to cold-stable microtubules. Each kinetochore was classified as 'attached' or 'not attached' depending on whether a microtubule fiber ended at the kinetochore. An average of 120 kinetochores was counted per cell, and seven cells were analyzed for each condition.

## Expression and purification of MBP-Knl1$^{138–168}$-H6

MBP-Knl1$^{138–168}$-H6 and MBP-Knl1$^{138–168}$-H6 constructs (MELT1) were obtained by sub-cloning into a pGEX vector backbone in which the coding sequence for GST was replaced with that for MBP. Expression was carried out in BL21 RIL strain at 25°C and by using 1 mM IPTG for 2.5 hr to induce expression. Cell pellets were re-suspended in three pellet volumes of 50 mM HEPES-NaOH pH 7.5, 250 mM KCl, 2 mM DTE, 10% glycerol, protease inhibitor mix (Serva). Cells were lysed by sonication, and the lysates were centrifuged at 100000×g for 1 hr at 4°C. Recombinant products were isolated from the lysate by using the HisTrap (GE Healthcare) column, followed by buffer exchange using a desalting column (GE Healthcare). Purified proteins were concentrated to about 3 mg/ml and frozen in liquid nitrogen.

## Expression and purification of H6-BubR1$^{362–431}$/Bub3 and H6-Bub1$^{209–270}$/Bub3 constructs

GST-BubR1$^{362–431}$/Bub3 and GST-Bub1$^{209–270}$/Bub3 constructs were obtained by sub-cloning the coding sequences for Bub3 and for the indicated segments of Bub1 into pFLMultiBac vector (*Trowitzsch et al., 2010*). Expression was carried out by infection of Tna38 insect cells (*Hashimoto et al., 2012*) for 72 hr at 27°C. Viruses of the constructs were generated as described (*Trowitzsch et al., 2010*). Insect cells were harvested by centrifugation at 1500 rpm for 30 min in a Sorvall RC 3BP+ (Thermo Scientific, Carlsbad, CA) centrifuge with Rotor H6000A, and the pellets were frozen in liquid nitrogen and stored at −80°C. 1 g of cell pellet was re-suspended in 10 ml lysis buffer

(50 mM Tris–HCl pH 8.0 or 8.5, 150 mM KCl, 2 mM DTE, DNAse, PMSF, protease inhibitors [Serva]). Cells were lysed by sonication and the lysate centrifuged at 100000×g for 1 hr at 4°C. The supernatant was filtered through Nalgene bottle-top filter. 1.5 ml of GSH bead slurry (GE Healthcare) was added to 50 ml of cleared lysate. After 1 hr at 4°C on a rotating wheel, the beads were recovered by centrifugation and washed with lysis buffer. Bead-bound complexes in 50 ml of lysis buffer were retrieved from the GSH beads by addition of GSH-Prescission protease (produced in house) for 14 hr at 4°C. Eluates were concentrated using Amicon concentrators (3 kDa cutoff), diluted with 20 mM Tris–HCl, 2 mM DTE to a final KCl concentration of 50 mM, and further purified using 1 ml HiTrap QFF column (GE Healthcare). Peak fractions were collected and concentrated down to a volume of 2 ml and further purified by size exclusion chromatography using the S75 16/60 column (GE Healthcare). Peak fractions were collected, concentrated to about 3 mg/ml, frozen in small aliquots in liquid nitrogen, and stored at −80°C. Sequences of loop swap constructs were as follows:

Bub1$^{209-270}$
RRVITISKSEYSVHSSLASKVDVEQVVMYCKEKLIRGESEFSFEELRAQKYNQRRKHEQWVN
Bub1$^{209-270}$-BubR1 loop
RRVITTRKPGKEEGDPLSKVDVEQVVMYCKEKLIRGESEFSFEELRAQKYNQRRKHEQWVN
BubR1$^{362-431}$
INHILSTRKPGKEEGDPLQRVQSHQQASEEKKEKMMYCKEKIYAGVGEFSFEEIRAEVFRKKLKEQREAE
BubR1$^{362-431}$-Bub1 loop
INHILSISKSEYSVHSSLAQRVQSHQQASEEKKEKMMYCKEKIYAGVGEFSFEEIRAEVFRKKLKEQREAE

## Expression and purification of H6-BubR1$^{1-571}$, H6-BubR1$^{1-571}$/Bub3, H6-BubR1$^{222-571}$/Bub3, H6-Bub1$^{1-409}$/Bub3, H6-Bub1$^{1-409}$/Bub3-TRX and H6-Bub1$^{1-280}$/Bub3

Sequences coding for H6-BubR1, H6-Bub1, and untagged Bub3 constructs were sub-cloned into pFLMultiBac vectors and baculoviruses were generated. Baculovirus expressing Bub3-TRX was generated by the Dortmund Protein Facility (DPF) using the pOPIN vector system {Berrow:2007cy}. Bub1/Bub3 and BubR1/Bub3 complexes were generated by co-infection and co-expression at 27°C for 72 hr. Insect cells were harvested by centrifugation at 1500 rpm for 30 min in a Sorvall RC 3BP+ (Thermo Scientific) centrifuge with Rotor H6000A, the pellets were frozen in liquid nitrogen and stored at −80°C. 1 g of cell pellet was re-suspended in 10 ml Lysis buffer (50 mM HEPES-KOH pH 7.5, 150 mM KCl, 15 mM imidazole, 2 mM DTE, 0.05% Tween20, PMSF, protease inhibitors [Serva]). Cells were lysed by sonication and centrifuged at 100000×g for 1 hr at 4°C. The supernatant was filtered through Nalgene bottle-top filter. The complexes were isolated from the cleared lysate on a 5-ml HisTrap (GE Healthcare) column. Peak fractions were pooled concentrated using Amicon concentrators and further purified in GF buffer (50 mM Hepes-KOH pH 7.5, 150 mM KCl, 2 mM DTE, 0.05% Tween20) by size exclusion chromatography using S200 16/60 column (GE Healthcare). Peak fractions were pooled, concentrated to typically 3 to 5 mg/ml, and frozen in liquid nitrogen.

## Expression and purification of H6-Bub1$^{271-409}$-MBP, H6-Bub1$^{209-409}$-MBP/Bub3, H6-TRX-BubR1$^{362-571}$, H6-TRX-BubR1$^{432-571}$

Sequences encoding H6-TRX-BubR1, H6-Bub1-MBP, and untagged Bub3 constructs were sub-cloned into pFLMultiBac vectors and baculoviruses were generated. All constructs, apart from H6-Bub1$^{209-409}$-MBP, which was co-expressed with untagged Bub3, were expressed individually in insect cells at 27°C for 72 hr. Insect cells were harvested by centrifugation at 1500 rpm for 30 min in a Sorvall RC 3BP+ (Thermo Scientific) centrifuge with Rotor H6000A, the pellets were frozen in liquid nitrogen and stored at −80°C. 1 g of cell pellets were re-suspended in 10 ml Lysis buffer (50 mM Tris–HCl pH 7.5, 150 mM KCl, 0.5 mM β-mercaptoethanol, 0.05% NP40, PMSF, protease inhibitors [Serva]). Cells were lysed by sonication and centrifuged at 100000×g for 1 hr at 4°C. The supernatant was filtered through Nalgene bottle-top filter. The complexes were isolated from the cleared lysate on a 5-ml TALON column (Clontech). Peak fractions were pooled, concentrated, and further purified in GF buffer (50 mM Tris–HCl pH 8.0, 150 mM KCl, 2 mM TCEP) by size exclusion chromatography on a S75 16/60 or S200 16/60 column. Peak fractions were pooled, concentrated to typically 3 to 5 mg/ml, frozen in small aliquots in liquid nitrogen, and stored at −80°C.

## Size exclusion chromatography mobility shift assay

Proteins tested for interactions were diluted to 15 µM in 150 µl reactions in GF buffer (50 mM Tris–HCl pH 8, 150 mM KCl, 1.5 mM TCEP, 0.05% Tween20) and incubated at 20–22°C for 1 hr on a rotating wheel. 100 µl of the resulting incubation was analyzed by size exclusion chromatography on a Superose6 10/300 column at a flow rate of 0.4 ml/min and collected in 300 µl fractions. Eluates were analyzed by SDS PAGE and Coomassie staining.

## In vitro binding assays

5 µg of MBP-Knl1$^{MELT1}$-H6 protein was phosphorylated with Mps1 kinase (TTK, Life technologies) in a 35 µl reaction in 12.5 mM Tris–HCl pH 7.5, 35 mM KCl, 10 mM MgCl$_2$, 0.5 mM EGTA, 0.005% Triton X-100, 2 mM TCEP, 0.5 µM Okadaic acid, at 30°C for 1 hr, 1200 rpm. BSA was added to a final concentration of 2 mg/ml, and the reaction was incubated with 20 µl Amilose resin (New England Biolabs, Ipswich, MA) for 1 hr at room temperature. The beads were washed with 50 mM HEPES-NaOH pH 7.5, 150 mM KCl, 0.05% Tween-20, 2 mM DTE. 20 µl Knl1-bound resin was then incubated in 50 µl of a 400 nM solution of prey proteins (e.g., H6-BubR1$^{362–431}$/Bub3 or H6-Bub1$^{209–270}$/Bub3 or loop swap constructs) in 50 mM HEPES-NaOH pH 7.5, 50 mM KCl, 0.05% Tween-20, 2 mM DTE, 10% glycerol, 4 mg/ml BSA, for 1 hr at room temperature. Unbound proteins were removed with 50 mM HEPES-NaOH pH 7.5, 150 mM KCl, 0.05% Tween-20, 2 mM DTE. 20 µl of the resin was boiled in 70 µl of sample buffer, separated on 10% gel, and blotted with anti-Bub3 (mouse monoclonal, BD, 1:1000) or anti-MBP (mouse monoclonal, New England Biolabs, 1:10000) antibodies.

## Mass spectrometry

Cells were adapted to Lys-0/Arg-0 (Light) medium or Lys-8/Arg-10 (Heavy) medium for 2 weeks. Cells were synchronized in mitosis by a 24-hr thymidine block, followed by a 14-hr treatment with nocodazole. After harvesting the mitotic population, cells were split in the presence of either 500 nM Reversine for 30 min or with DMSO as a control. LAP-BUB1 or LAP-KNL1 expression was induced for 24 hr using doxycycline and cells were harvested and mixed, followed by immunoprecipitation and mass spectrometry. Cells were lysed at 4°C in hypertonic lysis buffer (500 mM NaCl, 50 mM Tris–HCl [pH 7.6], 0.1% sodium deoxycholate, 1 mM DTT) including phosphatase inhibitors (1 mM sodium orthovanadate, 5 mM sodium fluoride, 1 mM β-glycerophosphate), sonicated, and LAP-tagged proteins were coupled to GFP-trap (ChromoTek) for 1 hr at 4°C. Purifications were washed three times with high-salt (2 M NaCl, 50 mM Tris–HCl (pH 7.6), 0.1% sodium deoxycholate, 1 mM DTT) and low-salt wash buffers (50 mM NaCl, 50 mM Tris–HCl (pH 7.6), 1 mM DTT) and subsequently eluted in 2 M Urea, 50 mM Tris-HCL (pH 7.6), 5 mM IAA. Samples were loaded on a C18 reverse phase column and ran on a nano-LC system coupled to a mass spectrometer (LTQ-Orbitrap Velos; Thermo Fisher Scientific) via a nanoscale LC interface (Proxeon Biosystems, now Thermo Fisher Scientific), as described in *Suijkerbuijk et al. (2012b)*.

## LacO experiments

U2OS LacO cells (a gift from S Janicki) were grown in DMEM supplemented with 8% FBS (Clontech), hygromycin (200 µg/ml), pen/strep (50 µg/ml), and L-glutamine (2 mM). Cells were transfected with the indicated constructs for 48 hr using Fugene HD according to the manufacturer's protocol. Asynchronously growing cells were arrested in prometaphase by the addition of nocodazole (830 nM) for 2–3 hr. Cells plated on 12-mm coverslips were fixed (with 3.7% paraformaldehyde, 0.1% Triton X-100, 100 mM Pipes, pH 6.8, 1 mM MgCl$_2$, and 5 mM EGTA) for 5–10 min. Coverslips were washed with PBS and blocked with 3% BSA in PBS for 1 hr, incubated with primary antibodies (GFP-booster [Chromotek], rabbit-anti-BUBR1 [Bethyl] and CREST/anti-centromere antibodies [Cortex Biochem, Inc.]) for 16 hr at 4°C, washed with PBS containing 0.1% Triton X-100, and incubated with secondary antibodies (goat-anti-rabbit Alexa Fluor 568 and goat anti-human Alexa Fluor 647) for an additional hour at room temperature. Coverslips were then washed, incubated with DAPI for 2 min, and mounted using antifade (ProLong; Molecular Probes, Eugene, OR). All images were acquired on a deconvolution system (DeltaVision RT; Applied Precision, part of GE Healthcare) with a 100×/1.40 NA U Plan S Apochromat objective (Olympus, Shinjuku, Tokyo, Japan) using softWoRx software (Applied Precision).

## Acknowledgements

We are grateful to Tim Bergbrede and the Dortmund Protein Facility for support with vector construction and to all members of the Musacchio laboratory for comments and discussions. KO is

enrolled in the International Max Planck Research School in Chemical Biology. AM acknowledges funding by the European Union's 7th Framework Program ERC agreement KINCON and the Integrated Project MitoSys.

## Additional information

### Funding

| Funder | Grant reference number | Author |
| --- | --- | --- |
| European Commission | Integrated Project MitoSys | Andrea Musacchio |
| European Research Council | Advanced Investigator Grant KINCON | Andrea Musacchio |

The funders had no role in study design, data collection and interpretation, or the decision to submit the work for publication.

### Author contributions

KO, IP, Conception and design, Acquisition of data, Analysis and interpretation of data, Drafting or revising the article; MV, VK, GJPLK, Analysis and interpretation of data, Drafting or revising the article, Contributed unpublished essential data or reagents; SM, Acquisition of data, Analysis and interpretation of data, Drafting or revising the article; IH, Acquisition of data, Analysis and interpretation of data; AM, Conception and design, Analysis and interpretation of data, Drafting or revising the article

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
