## [Decision Letter]

Thank you for sending your work entitled “Sub-functionalization of Bub1 and BubR1 paralogs from subtle sequence changes at two hetero-dimerization interfaces” for consideration at *eLife*. Your article has been favorably evaluated by Randy Schekman (Senior editor) and 3 reviewers, one of whom, Stephen Harrison, is a member of our Board of Reviewing Editors.

The Reviewing editor and the other reviewers discussed their comments before we reached this decision, and the Reviewing editor has assembled the following comments to help you prepare a revised submission.

Overlack and co-workers report a detailed and systematic analysis of how Bub1 and BubR1 localize to kinetochores and how these proteins regulate cell division and cell cycle progression in human cells. Previous efforts to distinguish the functions of Bub1 and BubR1, two closely related proteins, have yielded confusing results. Overlack and co-workers have carried out a series of well-designed experiments that lead to a plausible model for the function of these proteins in dividing cells. In this model, Bub1 binds Bub3 to form a complex that recruits the BubR1/Bub3 complex. By examining dissecting kinetochore recruitment of BubR1, the authors propose that its kinetochore recruitment may not be critical for its checkpoint function but rather for establishing proper chromosome-microtubule attachments.

All three reviewers are positive and support publication. One made the following detailed comments and requests, enumerated by figure.

Figure 1: How good is the Bub1 (and BubR1) RNAi in these studies? Some quantitation should be shown.

C/D: Was there any sign of Mad1 in the mass-spec? This is relevant to the model in Figure 8.

Figure 2: There is an important point here: are all MELT repeats created equal? Presumably not, or the array of repeats on KNL1 would not be so complicated in so many organisms. The issue here is that these KNL1 pull downs are only done with one of the MELT motifs (KNL1 138-168). How representative is this one? Can BubR1-Bub3 bind much more tightly to other MELT repeats or when multiple repeats are present?

There are many options possible here; the authors should at least address this point. How far can their findings in Figure 2 be generalized to other MELT motifs on KNL1?

Figure 4: The expression levels for all the different GFP-Bub1 fusions should be demonstrated by anti-GFP western. For example the 1-284 fragment gives very bright GFP-Bub1 staining at kinetochores. Is that due to expression levels, or because the TPR region significantly enhances kinetochore binding? This distinction is important, as the manuscript (e.g. in the Results section) is rather dismissive about the role of the Bub1-TPR in kinetochore targeting. It does seem clear that the TPRs have little role in the Bub1-BubR1 interaction.

Figure 6: The tethering experiments are not very convincing as presented and should be improved. The merged images in panel H both look rather green on the reviewer's screen/paper. The co-localisation should be quantitated somehow. Does a mutation/deletion in the Bub1 271-409 region abolish BubR1 recruitment? Can tethered BubR1 recruit Bub1 in a Bub3 dependent fashion?

Figure 7: The observation that BubR1-D432-484 can bind APC/C and provide SAC delay yet fails to target to kinetochores is quite striking. It also fails to bind PP2A, so an additional effect on SAC silencing makes the delay harder to interpret. The authors should state more clearly that MCC-APC/C is being generated efficiently, independently of the kinetochore targeting of BubR1. This interpretation is relevant for the model in Figure 8.

Figure 8: This is a KNL1 based model, which is fine for wild-type cells (but see below). The representation of KNL1 is unclear: what do the purple oval and black lines represent? Is it drawn C-N here? The TPR interactions also need explaining in the model: what are they binding to on KNL1?

Finally, although the kinetochore template can generate an MCC copy, Figure 7 shows that robust MCC-APC/C complex can be formed by BubR1-D432-484 which the authors have shown does not bind Bub1 nor go to kinetochores. How is that MCC generated?

The Reviewing editor found the text very rough going. Apparently it is fine for experts (the other two reviewers had less trouble), but even a semi-expert got badly lost. The main problem is unnecessarily turgid and overwrought prose. Keep in mind the following outline: 1) The goal is to understand the differential roles of paralogs Bub1 and BubR1. (Therefore eliminate the first paragraph, or at least move it to the Discussion. This is a nice paper about Bub1 and BubR1, not a disquisition on strategies of evolution.)

2) Ectopically expressed Bub3 binding segment (also known as GLEBS) of BubR1 does not associate with kinetochores, even though the paralogous segment of Bub1 does. (Is this result completely new, or are there published experiments that come to the same conclusion?)

3) “Loop swap” experiments show that the loop between β1 and β2 of the Bub3 binding segment accounts for at least some of the differential properties of the two paralogs. The Bub1 loop promotes interaction of Bub1:Bub3 with a phosphorylated MELT-repeat segment from Knl1 (in vitro); the BubR1 loop does not. (These are “pull-down” experiments, which tend to have an exaggeratedly all-or-none character, but the results seem OK to the unpracticed eye. But see the comment on Figure 2, above. Interaction properties of multiple repeats are not the sum of their individual properties, as Rosen and others have emphasized.)

4) Kinetochore localization experiments in HeLa cells give results consistent with the conclusion in #3.

5) BubR1 with a Bub1 loop does not associate with MCC or APC. These are IP experiments and seem OK.

6) Bub1 and BubR1 interact with each other. The “pseudo-symmetric” (paralogous pairing need not be pseudo-symmetric—is there evidence for a genuinely pseudo-symmetric contact?) interaction requires the Bub3-binding motif and a putatively helical bit C-terminal to it. The experiments that establish this point are described with such an elaborate layering of deletions that the Reviewing editor found the logic very hard to follow. Please try to condense the text from the subsection “A minimal BubR1-binding region of Bub1” until “Functional relevance to kinetochore recruitment of BubR1” in the Results section, perhaps to half the current length. Would it help to make Figure 6 both more prominent and more elegant?

7) The two expert reviewers believe that the section on functional relevance is important, but it reads to the less expert reader a bit like a set of flailing attempts, no one of which established an unambiguous conclusion. Again, try to condense and get to the point.

In summary, the reviewers will welcome a revised version that responds to each of the seven points above and ideally one that is a bit less florid and therefore more readable. The one experimental request concerns Figure 6 (see above). The other points are either clarifications or requests for explicit mention of data the authors are likely already to have (e.g., information on the extent of knockdown in the experiments in Figure 1).

---

## [Author Response]

Figure 1*: How good is the Bub1 (and BubR1) RNAi in these studies? Some quantitation should be shown*.

We have now added two panels to Figure 1—figure supplement 1 (panels A and B) reporting quantification of RNAi experiments.

*C/D: Was there any sign of Mad1 in the mass-spec? This is relevant to the model in*
Figure 8.

Studies of the interaction of Mad1 with kinetochores in human cells have been puzzling. For reasons that are unclear to us and to the rest of the community, it proved very difficult to “lock” Mad1 in a stable complex with kinetochores in human cells, despite FRAP data showing that Mad1 interacts quite stably with kinetochores in prometaphase. We speculate that such behavior reflects a cooperative binding mechanism that is strongly weakened upon cell lysis and solubilization of kinetochores, possibly as a consequence of dephosphorylation of the relevant receptor. Indeed, we did not identify Mad1 in the IPs. However, an interaction of Mad1 with Bub1 was reported for *S. cerevisiae* and *C. elegans*, and in addition we recently reported that Mad1 can be identified in IPs of the first MELT region of Knl1, and that Bub1 is also present in such IPs (26). For all these reasons, we contend that the model in Figure 8 is quite likely to be accurate. We now discuss these facts explicitly at the end of the manuscript.

Figure 2*: There is an important point here: are all MELT repeats created equal? Presumably not, or the array of repeats on KNL1 would not be so complicated in so many organisms. The issue here is that these KNL1 pull downs are only done with one of the MELT motifs (KNL1 138-168). How representative is this one? Can BubR1-Bub3 bind much more tightly to other MELT repeats or when multiple repeats are present?*

*There are many options possible here; the authors should at least address this point. How far can their findings in*
Figure 2
*be generalized to other MELT motifs on KNL1?*

We have previously shown that the isolated first MELT (MELT1) repeat of Knl1 is extremely efficient and can support a robust spindle checkpoint response (63; 26). In a study that is now in press in Molecular Cell, Vleugel and colleagues (laboratory of G. Kops, co-authors on this manuscript) analyzed this question systematically. Their main conclusion is that MELT 1 is among the three best MELT repeats, as far as SAC signaling and the interaction with Bub1 and BubR1 are concerned. Because we show that full length BubR1 carrying the loop swap mutation is recruited to kinetochores in a Bub1-independent manner in the presence of endogenous Knl1, and that full length Bub1 carrying the loop swap mutation cannot be recruited to Knl1 any longer, our experiments with MELT1 in vitro clearly support a general behavior, not one that is limited to MELT1.

Figure 4*: The expression levels for all the different GFP-Bub1 fusions should be demonstrated by anti-GFP western. For example the 1-284 fragment gives very bright GFP-Bub1 staining at kinetochores. Is that due to expression levels, or because the TPR region significantly enhances kinetochore binding? This distinction is important, as the manuscript (e.g. in the Results section) is rather dismissive about the role of the Bub1-TPR in kinetochore targeting. It does seem clear that the TPRs have little role in the Bub1-BubR1 interaction*.

In two previous papers (27; 26) we characterized in detail the role of the TPR domain of Bub1 and demonstrated that the TPR, which interacts with so-called KI motifs of human Knl1, is dispensable for kinetochore localization but increases the binding affinity of Bub1 for kinetochores, a conclusion based on a non-equilibrium assay such as immune-precipitation. (In those papers, we had already included expression controls for our constructs.) We have added a short reference to our previous work and to the role of KI motifs in the revised version of the manuscript. We include here for the reviewers’ view a western blot documenting the expression levels of the different constructs (Figure 9). The Bub1^1 -788^ construct is expressed at very low levels, but this construct has no relevance for any of our conclusions because all residues C-terminal to 270 are dispensable for kinetochore localization of Bub1.Author response image 1.anti-GFP Western expression levels of GFP-Bub1 constructs

Figure 6*: The tethering experiments are not very convincing as presented and should be improved. The merged images in panel H both look rather green on the reviewer's screen/paper. The co-localisation should be quantitated somehow*. *Does a mutation/deletion in the Bub1 271-409 region abolish BubR1 recruitment? Can tethered BubR1 recruit Bub1 in a Bub3 dependent fashion?*

We appreciate the potential interest of the experiments proposed by the reviewer, although we also note that they are confirmatory given that we have demonstrated the interaction already in cells and with purified proteins in vitro. Regretfully the person in charge of these experiments, Dr. Mathijs Vleugel, a member of the laboratory of Prof. Kops, has now moved to the laboratory of Prof. Marileen Dogterom for post-doctoral training. As it would take a relatively long time to train a new person to carry out these experiments, we hope that the reviewers will accept our request not to perform them. We have included a better image and a quantitation of these experiments.

Figure 7*: The observation that BubR1-D432-484 can bind APC/C and provide SAC delay yet fails to target to kinetochores is quite striking. It also fails to bind PP2A, so an additional effect on SAC silencing makes the delay harder to interpret. The authors should state more clearly that MCC-APC/C is being generated efficiently, independently of the kinetochore targeting of BubR1. This interpretation is relevant for the model in*
Figure 8.

We now write (in the third paragraph of the Discussion section): The apparent checkpoint proficiency of BubR1^Δ432-484^ may indicate that BubR1 localization to kinetochores is not essential for checkpoint function, as it implies that substantial amounts of MCC can be generated even when BubR1/Bub3 cannot be recruited to kinetochores.

Figure 8*: This is a KNL1 based model, which is fine for wild-type cells (but see below). The representation of KNL1 is unclear*: *what do the purple oval and black lines represent? Is it drawn C-N here? The TPR interactions also need explaining in the model: what are they binding to on KNL1?*

*Finally, although the kinetochore template can generate an MCC copy,*
Figure 7
*shows that robust MCC-APC/C complex can be formed by BubR1-D432-484 which the authors have shown does not bind Bub1 nor go to kinetochores. How is that MCC generated?*

We have now added “N” and “C” to clarify the orientation of Knl1 in these images. We now also clarify in the text and in the legend to Figure 8 that the TPR motifs bind to KI motifs on Knl1. We omit these additional elements from the figure to avoid charging it excessively.

The reviewer also argues that even if the template generated an MCC copy, this reaction may not be functionally relevant because the copy can be generated away from kinetochores. It is important to clarify that in Figure 8 we don’t want to imply that MCC can only be formed through a template at the kinetochore. The figure only tries to convey the fact that there are two structures, one at kinetochores and one in the cytosol, whose relation appears to be one of template and copy. We think that it is important to point out this relationship because it summarizes the significance of the divergence of Bub1 and BubR1 and of the conversion of Mad2. Admittedly, many of the details of the interactions depicted on this figure remain obscure. We have tried to clarify this in the text.

*The Reviewing editor found the text very rough going. Apparently it is fine for experts (the other two reviewers had less trouble), but even a semi-expert got badly lost. The main problem is unnecessarily turgid and overwrought prose. Keep in mind the following outline: 1) The goal is to understand the differential roles of paralogs Bub1 and BubR1. (Therefore eliminate the first paragraph, or at least move it to the Discussion. This is a nice paper about Bub1 and BubR1, not a disquisition on strategies of evolution*.*)*

We moved the first paragraph to the beginning of the Discussion. We have considerably shortened and simplified the Introduction, which is now more to the point.

2) Ectopically expressed Bub3 binding segment (also known as GLEBS) of BubR1 does not associate with kinetochores, even though the paralogous segment of Bub1 does. (Is this result completely new, or are there published experiments that come to the same conclusion?)

Yes, the result is completely new, to the best of our knowledge.

*3) “Loop swap” experiments show that the loop between β1 and β2 of the Bub3 binding segment accounts for at least some of the differential properties of the two paralogs. The Bub1 loop promotes interaction of Bub1:Bub3 with a phosphorylated MELT-repeat segment from Knl1 (in vitro); the BubR1 loop does not. (These are “pull-down” experiments, which tend to have an exaggeratedly all-or-none character, but the results seem OK to the unpracticed eye. But see the comment on*
Figure 2*, above. Interaction properties of multiple repeats are not the sum of their individual properties, as Rosen and others have emphasized*.*)*

As clarified above in our answer to the comment on Figure 2, the results in vitro are fully validated in cells, i.e. in the presence of endogenous Knl1, from which we gather that the results in vitro are reliable.

4) Kinetochore localization experiments in HeLa cells give results consistent with the conclusion in #3).

Agreed.

*5) BubR1 with a Bub1 loop does not associate with MCC or APC. These are IP experiments and seem OK*.

We have carried out these experiments several times and trust these results, even if we cannot yet explain them in full.

*6) Bub1 and BubR1 interact with each other. The “pseudo-symmetric” (paralogous pairing need not be pseudo-symmetric—is there evidence for a genuinely pseudo-symmetric contact?) interaction requires the Bub3-binding motif and a putatively helical bit C-terminal to it. The experiments that establish this point are described with such an elaborate layering of deletions that the Reviewing editor found the logic very hard to follow. Please try to condense the text from the subsection “A minimal BubR1-binding region of Bub1” until “Functional relevance to kinetochore recruitment of BubR1” in the Results section, perhaps to half the current length. Would it help to make*
Figure 6
*both more prominent and more elegant?*

We have tried to simplify the description of the binding experiments to the best of our ability. The style we have used to depict constructs in Figure 6 is the same we have used throughout the manuscript, and therefore opted not to change it, but we have harmonized the two parts of the table and corrected some small problems with numbering and length of the constructs. We now discuss the summary table before (i.e. as panel B of Figure 6) the description of the experiments with the BubR1^Δ432-484^ deletion mutant (now panels C-D, formerly, B-C). We hope the reviewers will find the revised description of the binding experiments easier to understand.

*7) The two expert reviewers believe that the section on functional relevance is important, but it reads to the less expert reader a bit like a set of flailing attempts, no one of which established an unambiguous conclusion. Again, try to condense and get to the point*.

We think that part of the reason why our conclusions may seem “weak” is that BubR1^Δ432-484^, the deletion mutant that cannot be recruited to kinetochores, is checkpoint proficient. It was important to discuss the behavior of this mutant, as well as its possible interpretations. We have tried to improve the discussion of this mutant, which, we realize, is complex, as it entails the interaction with PP2A, the possibility that the deletion removes an important regulatory intramolecular interaction, etc. With this word of caution, the work carries a whole lot of new, surprising, and very relevant functional data, the most relevant of which is that it allows for the first time to understand how Bub1 and BubR1 are recruited to kinetochores. We also demonstrate that the “loop” regions of Bub1 and BubR1 impart very different behaviors to Bub3, and that they cannot be replaced without fundamentally altering the functions of these proteins, a quite striking manifestation of sub-functionalization. We show that Bub1 and BubR1 interact directly. And we demonstrate that kinetochore-microtubule attachment is crucially dependent on kinetochore localization of BubR1. The latter is not a novel conclusion, but it is obtained here with a “separation of function” mutant of BubR1 that is checkpoint proficient. For all these reasons, we think that this paper will be greatly influential.